# DIVERGENCE-FREE NEURAL NETWORKS WITH APPLICATION TO IMAGE DENOISING

**Sébastien Herbreteau**
Univ Rennes, Ensai, CNRS, CREST—UMR 9194
F-35000 Rennes, France
sebastien.herbreteau@ensai.fr

**Etienne Meunier**
Centre Inria de Paris
Paris, France
etienne.meunier@inria.fr

## ABSTRACT

We introduce a resource-efficient neural network architecture with zero divergence by design, adapted for high-dimensional problems. Our method is directly applicable to image denoising, for which divergence-free estimators are particularly well-suited for self-supervised learning, in accordance with Stein's unbiased risk estimation theory. Comparisons of our parameterization on popular denoising datasets demonstrate that it retains sufficient expressivity to remain competitive with other divergence-based approaches, while outperforming its counterparts when the noise level is unknown and varies across the training data.

## 1 INTRODUCTION

The divergence is a scalar quantity that measures the rate at which a vector field "flows out" of an infinitesimal region of space. Formally, for a weakly differentiable function $f : \mathbb{R}^n \to \mathbb{R}^n$, the divergence at point $\boldsymbol{y} \in \mathbb{R}^n$ is defined as the trace of the Jacobian matrix $\boldsymbol{J}_f(\boldsymbol{y})$:

$$\operatorname{div} f(\boldsymbol{y}) \triangleq \operatorname{tr}(\boldsymbol{J}_f(\boldsymbol{y})) = \sum_{i=1}^{n} \frac{\partial f_i}{\partial y_i}(\boldsymbol{y}) \, . \tag{1}$$

In the special case where the divergence is zero everywhere, the vector field is said to be divergence-free or solenoidal, indicating an incompressible flow. One of the most famous example is without doubt the magnetic field, which, according to Maxwell's equations, has zero divergence (Maxwell, 1873). Learning divergence-free vector fields is of particular interest at the interface of physics and machine learning (Richter-Powell et al., 2022; Raissi et al., 2017a), as such fields naturally emerge in systems governed by fundamental conservation laws. Parameterizations for learning often exploit the fact that, in $\mathbb{R}^3$, the curl of any vector field is divergence-free (Morita, 2001), or, more generally, draw on its multidimensional extension via differential forms (Cartan, 1899; Richter-Powell et al., 2022). As long as the target functions remain low-dimensional, training can be performed efficiently with the help of an automatic differentiation engine (Paszke et al., 2019) that powers the computation of partial derivatives. However, scaling challenges arise quickly as the dimensionality increases (Richter-Powell et al., 2022).

In this paper, we establish a representer theorem for divergence-free vector fields, based on structured combinations of conservative fields. Building on this result, and incorporating sparsity constraints, we show how this representation can inform neural network parameterizations that remain resource-efficient, thereby ensuring computational tractability in high dimension. With application to image denoising, for which divergence-free estimators are particularly well-suited for self-supervised learning, in accordance with Stein's unbiased risk estimation theory Stein (1981), we propose a methodology to construct low-overhead network architectures that have zero divergence by design and which are adapted to image processing tasks. We demonstrate their competitiveness in comparison to other divergence-based approaches (Batson & Royer, 2019; Tachella et al., 2025a; Soltanayev & Chun, 2018) for the removal of Gaussian noise without clean data.

In summary, the contributions of our work are as follows:

1. The establishment of a representer theorem for divergence-free fields, on which we build to construct neural network architectures that have zero divergence by design.

2. A theoretical framework for analyzing self-supervised image denoising methods grounded in the principle of constant divergence.

3. The demonstration of the competitiveness of our approach in comparison with other constraint-based approaches, particularly when the noise level is unknown and varies across the training data.

## 2 RELATED WORK

Divergence-free networks are particularly studied within physics-informed machine learning and related scientific modeling tasks, which integrate physical laws into the training of neural networks to solve partial differential equations. Notably, enforcing incompressibility constraints is often important—especially in fluid dynamics, where velocity fields are required to be divergence-free.

A common approach employs soft constraints by adding penalty terms to the loss function that encourage the predicted fields to be divergence-free (Raissi et al., 2017b; Mao et al., 2020; Jin et al., 2021). Although such penalty-based methods are straightforward to implement, they do not guarantee strict satisfaction of the incompressibility condition, and residual divergence can remain in some cases, particularly for complex or high-dimensional problems.

To overcome these limitations, recent works have explored hard constraints that enforce divergence-free properties by construction through network architecture or parameterization. For example, in Raissi et al. (2017a), a stream function formulation is used in 2D to represent the velocity field as derivatives of a scalar network output, which is analytically divergence-free. Extending this idea to the multidimensional case, Richter-Powell et al. (2022) designed networks that directly encode conservation laws—including divergence-free constraints—thereby allowing modeling of flow fields and advected quantities without explicit divergence penalties. Nonetheless, scaling these models proves challenging due to their heavy reliance on automatic differentiation. For example, the vector-field parameterization proposed by Richter-Powell et al. (2022) requires computing a Jacobian matrix, which becomes intractable as the dimension grows.

## 3 CONSTRAINT-BASED APPROACHES FOR SELF-SUPERVISED DENOISING

We focus on denoising problems under the assumption of additive white Gaussian noise (AWGN):

$$\boldsymbol{y} = \boldsymbol{x} + \sigma\boldsymbol{\epsilon}, \tag{2}$$

where $\boldsymbol{y} \in \mathbb{R}^n$ is the noisy observation, $\boldsymbol{x} \in \mathbb{R}^n$ is the underlying noise-free signal drawn from a distribution $p_{\boldsymbol{x}}$, $\boldsymbol{\epsilon} \sim \mathcal{N}(\boldsymbol{0}, \boldsymbol{I}_n)$ is a standard Gaussian noise vector, and $\sigma > 0$ is the noise level, drawn from a distribution $p_\sigma$. Provided that a sufficiently large dataset of noise-free signals $\boldsymbol{x}$ is available, problem (2) is traditionally tackled in a supervised manner by solving:

$$\arg\min_f \mathbb{E}_{\boldsymbol{x},\boldsymbol{y}}\|f(\boldsymbol{y}) - \boldsymbol{x}\|_2^2, \tag{3}$$

that is, by finding the minimum mean square error (MMSE) estimator, which we denote $f^{\mathrm{MMSE}}$. Interestingly, $f^{\mathrm{MMSE}}$ admits a closed-form expression whenever the noise level $\sigma$ is the same for all noisy observations $\boldsymbol{y}$ (i.e., $p_\sigma$ is a Dirac) which is given by Tweedie's formula (Efron, 2011): $f^{\mathrm{MMSE}}(\boldsymbol{y}) = \boldsymbol{y} + \sigma^2 \nabla \log p_{\boldsymbol{y}}(\boldsymbol{y})$. In this latter expression, the optimal estimator $f_{\mathrm{MMSE}}$ depends solely on the score of the distribution of the noisy data $\nabla \log p_{\boldsymbol{y}}(\boldsymbol{y})$. Accordingly, this formulation indicates that Gaussian denoising may be effectively addressed even in the absence of ground-truth data $\boldsymbol{x}$ for training.

### 3.1 STEIN'S UNBIASED RISK ESTIMATE

Among all the methods proposed in the literature for tackling self-supervised denoising, the approaches grounded in Stein's Unbiased Risk Estimate (SURE) (Stein, 1981) hold a prominent place. This latter establishes a remarkable identity involving the divergence operator:

$$\mathbb{E}_{\boldsymbol{x},\boldsymbol{y}}\|f(\boldsymbol{y}) - \boldsymbol{x}\|_2^2 = \mathbb{E}_{\boldsymbol{y},\sigma}\left[-n\sigma^2 + \|f(\boldsymbol{y}) - \boldsymbol{y}\|_2^2 + 2\sigma^2\,\mathrm{div}\,f(\boldsymbol{y})\right], \tag{4}$$

provided that $f$ belongs to $\mathcal{L}^1$, the space of weakly differentiable functions, and under the assumption that $\mathbb{E}_{\boldsymbol{y}|\boldsymbol{x}}|f_i(\boldsymbol{y})|$ is bounded. This result is particularly powerful, as it reveals that the mean square error can be reformulated to depend solely on noisy observations and noise levels.

**When the noise level $\sigma$ is known:** Assuming that $\sigma$ is known for every noisy training signal $\boldsymbol{y}$, equation (4) allows (3) to be rewritten in the following equivalent self-supervised form:

$$\arg\min_f \ \mathbb{E}_{\boldsymbol{y},\sigma} \left[ \|f(\boldsymbol{y}) - \boldsymbol{y}\|_2^2 + 2\sigma^2 \operatorname{div} f(\boldsymbol{y}) \right], \tag{5}$$

where the divergence term acts as a form of regularization, preventing the model from simply learning the identity. Many traditional image denoisers—whose divergence admits a closed-form expression—are in fact rooted in SURE (Blu & Luisier, 2007; Van De Ville & Kocher, 2009; Wang & Morel, 2013), even if this connection is not made explicit in some cases (Dabov et al., 2007; Lebrun et al., 2013), as shown by Herbreteau & Kervrann (2025). However, when the estimator $f$ is considerably more complex, such as a deep neural network, its divergence is generally intractable to compute exactly. A popular method is then to employ a Monte Carlo approximation of the divergence, grounded in the following result (Ramani et al., 2008):

$$\operatorname{div} f(\boldsymbol{y}) = \lim_{\tau \to 0} \mathbb{E}_{\boldsymbol{h} \sim \mathcal{N}(\boldsymbol{0},\boldsymbol{I})} \left[ \boldsymbol{h}^\top \frac{f(\boldsymbol{y} + \tau \boldsymbol{h}) - f(\boldsymbol{y})}{\tau} \right], \tag{6}$$

provided that $f$ admits a well-defined second-order Taylor expansion (if not, this is still valid in the weak sense provided that $f$ is tempered, which is the case for networks with piecewise differentiable activation functions as shown by Soltanayev & Chun (2018)). In practice, a single realization $\boldsymbol{h}$ from the standard normal distribution $\mathcal{N}(\boldsymbol{0},\boldsymbol{I})$ is used for approximating the divergence and $\tau$ is chosen as a small constant. In total, only two evaluations of the function $f$ are necessary to estimate its divergence with this method. In a deep learning setting, Soltanayev & Chun (2018); Chen et al. (2022) leveraged this Monte Carlo approximation in combination to the SURE loss (5) to train neural networks on datasets composed only of noisy observations $\boldsymbol{y}$, leading to the **MC-SURE** approach. While they achieved performance close to that of the MMSE estimator, a slight gap remains, partly due to approximation errors in the divergence term.

**When the noise level $\sigma$ is unknown:** In more realistic settings, one typically has access only to a dataset composed of noisy signals, with no information about the underlying noise level $\sigma$ for each observation $\boldsymbol{y}$. Although several methods have been proposed for ad hoc noise estimation (Chen et al., 2015; Pyatykh et al., 2013; Foi et al., 2008), a recent line of research (Tachella et al., 2025a; Batson & Royer, 2019; Krull et al., 2019) seeks to circumvent the need for estimating $\sigma$ altogether. This is achieved by restricting the estimator $f$ to belong to a constrained set $\mathcal{S}$ such that

$$\exists \lambda \in \mathbb{R}, \forall f \in \mathcal{S}, \quad \mathbb{E}_{\boldsymbol{y},\sigma} \left[ \sigma^2 \operatorname{div} f(\boldsymbol{y}) \right] = \lambda, \tag{7}$$

thereby effectively removing this term from the optimization objective (5). The final objective then consists in minimizing the measurement consistency under constraint:

$$\arg\min_f \ \mathbb{E}_{\boldsymbol{y}} \|f(\boldsymbol{y}) - \boldsymbol{y}\|_2^2 \quad \text{s.t.} \quad f \in \mathcal{S}. \tag{8}$$

In what follows, we describe two distinct types of constraints proposed in the literature for tackling self-supervised denoising with **unknown** noise levels, propose an alternative constraint set and then study their shared properties.

**Remark** In addition to the constraint-based approaches studied in this paper, we also mention, for completeness, the approaches that directly utilize the score function $\nabla \log p_{\boldsymbol{y}}(\boldsymbol{y})$ in Tweedie's formula, as proposed in Kim & Ye (2021); Kim et al. (2022); Xie et al. (2023), all of which depend on the estimation technique introduced by Lim et al. (2020). Moreover, Noise2Noise-like (Lehtinen et al., 2018) data augmentation techniques were also proposed (Pang et al., 2021; Huang et al., 2021; Wang et al., 2022; Mansour & Heckel, 2023) as an alternative to SURE.

### 3.2 Blind-spot estimators

A radical way to achieve (7) is to impose that each component function $f_i$ does not depend on $y_i$. Under this constraint, $f$ is trivially divergence-free by construction since $\frac{\partial f_i}{\partial y_i} = 0$. This idea dates back to Efron (2004) and lies at the core of the **Noise2Self** approach (Batson & Royer, 2019) and its variants (Krull et al., 2019; Laine et al., 2019), in which a so-called "blind-spot" network architecture

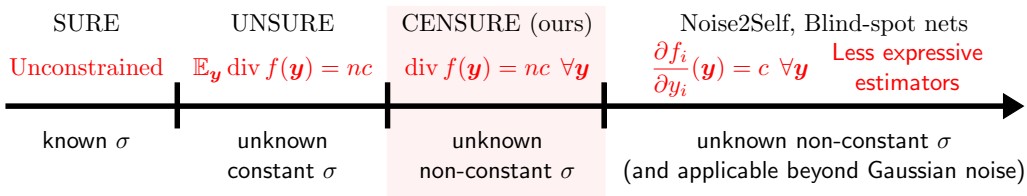

Figure 1: Balancing expressive power in constraint-based self-supervised Gaussian denoising. When the noise model is fully specified, SURE yields the most expressive estimators, whose performance matches theoretically that of supervised methods. As assumptions on the noise are gradually relaxed, however, the learned estimators must reduce their expressivity to avoid overfitting the noise. In this work, we introduce *divergence-constant estimators*, a family that preserves much of the expressivity lost in blind-spot designs, yet still allows training even when the noise level $\sigma$ is unknown and varies across the data.

is employed. From a broader perspective, this constraint can be generalized by restricting $f$ to the space

$$\mathcal{S}_{\text{BS}}^c = \{ f \in \mathcal{L}^1(\mathbb{R}^n, \mathbb{R}^n) \ : \ \forall \boldsymbol{y} \in \mathbb{R}^n, \frac{\partial f_i}{\partial y_i}(\boldsymbol{y}) = c \}, \tag{9}$$

which still satisfies (7) as long as $c \in \mathbb{R}$ is a constant fixed in advance. In the case $c = 0$, the solution of (8) under blind-spot constraints is given by $f_i^{\text{BS}}(\boldsymbol{y}) = \mathbb{E}\{y_i|\boldsymbol{y}_{-i}\} = \mathbb{E}\{x_i|\boldsymbol{y}_{-i}\}$, where $\boldsymbol{y}_{-i}$ refers to the vector obtained by excluding the $i$th entry.

The strength of blind-spot constraints lies actually in their versatility: they can handle a wide range of noise types, specifically those that are zero-mean and spatially independent, of which (2) is a prime example, without precisely knowing the noise distribution. However, this flexibility comes at a significant performance cost. A blind-spot architecture is indeed inherently less expressive than a classic one, especially since $y_i$ is usually highly informative about $x_i$, and tends to introduce checkerboard artifacts (Höck et al., 2022).

### 3.3 UNSURE ESTIMATORS

In the particular case where every noisy observation $\boldsymbol{y}$ is corrupted by the same noise level $\sigma$ (*i.e.*, $p_\sigma$ is a Dirac), Tachella et al. (2025a) proposed a softened version of the constraint (9) imposing only that the estimator has *zero expected divergence*, that is, $\mathbb{E}_{\boldsymbol{y}} \operatorname{div} f(\boldsymbol{y}) = 0$. This relaxation retains the validity of (7) while offering increased expressivity, since any blind-spot estimator automatically satisfies the zero–expected-divergence condition. Extending it to the constant case, this alternative constraint forces $f$ to belong to the space

$$\mathcal{S}_{\text{CED}}^c = \{ f \in \mathcal{L}^1(\mathbb{R}^n, \mathbb{R}^n) \ : \ \mathbb{E}_{\boldsymbol{y}} \operatorname{div} f(\boldsymbol{y}) = nc \}. \tag{10}$$

In the case $c = 0$, a closed-form solution of the optimization problem (8) was established, namely $f^{\text{ZED}}(\boldsymbol{y}) = \boldsymbol{y} + \hat{\eta} \nabla \log p_{\boldsymbol{y}}(\boldsymbol{y})$, with $\hat{\eta} = (\mathbb{E}_{\boldsymbol{y}} \frac{1}{n} \|\nabla \log p_{\boldsymbol{y}}(\boldsymbol{y})\|_2^2)^{-1}$.

An important difference with the blind-spot approach lies in how the optimization is carried out in practice. Indeed, unlike blind-spot approaches—where the constraint is enforced directly through the design of $f$—the so-called **UNSURE** approach seeks a saddle point of the Lagrangian by formulating the problem as a min–max optimization which is solved by alternating gradient-descent-ascent (Arrow et al., 1958; Platt & Barr, 1987). However, such an optimization method comes with several caveats. First, constraint satisfaction is not guaranteed in practice; only the penalty term associated with violations is minimized. Second, the outcome is highly sensitive to the choice of learning-rate pair for gradient-descent-ascent, which controls the trade-off between the objective and the constraint, and an inappropriate choice can lead to instabilities or oscillatory dynamics during training (Platt & Barr, 1987; Gallego-Posada et al., 2022). Finally, in this setting, the divergence term is estimated via a Monte Carlo approximation based on (6) using a limited number of samples, which can further degrade the accuracy of the optimization.

### 3.4 PROPOSED ALTERNATIVE: CENSURE ESTIMATORS

In this work, we propose to study the set of weakly differentiable vector fields on $\mathbb{R}^n$ with constant (normalized) divergence $c \in \mathbb{R}$, denoted by

$$\mathcal{S}_{\mathrm{DC}}^c = \{f \in \mathcal{L}^1(\mathbb{R}^n, \mathbb{R}^n) \ : \ \forall \boldsymbol{y} \in \mathbb{R}^n, \mathrm{div}\, f(\boldsymbol{y}) = nc\}, \tag{11}$$

*de facto* introducing an intermediate constraint set lying between the blind-spot constraint set and the much looser expected divergence constraint one: $\mathcal{S}_{\mathrm{BS}}^c \subset \mathcal{S}_{\mathrm{DC}}^c \subset \mathcal{S}_{\mathrm{CED}}^c$. We emphasize that all inclusions are strict, with in particular the possibility for $f \in \mathcal{S}_{\mathrm{DC}}^0$ to have each of its component function $f_i$ to depend on $y_i$, which is excluded for a function in $\mathcal{S}_{\mathrm{BS}}^0$ (see Fig. 6 in Appendix). We postpone the description of the way we construct in practice such divergence-constant estimators to the next section. Let us simply note that, similarly to existing alternatives, divergence-constant mappings are of particular interest in self-supervised denoising since the condition (7) is always satisfied, regardless of whether the noise level $\sigma$ is constant across observations $\boldsymbol{y}$ or varies from one observation to another. This is a major difference with the **UNSURE** approach which cannot, *a priori*, be used when the noise level varies across the noisy observations. Indeed, in full generality, $\mathbb{E}_{\boldsymbol{y},\sigma}[\sigma^2 \mathrm{div}\, f(\boldsymbol{y})] \neq \mathbb{E}_\sigma[\sigma^2]\mathbb{E}_{\boldsymbol{y}}[\mathrm{div}\, f(\boldsymbol{y})]$ because $\sigma^2$ is not independent of $\mathrm{div}\, f(\boldsymbol{y})$, since $\boldsymbol{y}$ itself depends on $\sigma$. Figure 1 illustrates the expressivity trade-off that emerges in self-supervised denoising. Our proposed approach, termed **CENSURE** (Concealed and Erratic Noise level with Stein's Unbiased Risk Estimate), bridges the gap between constant expected divergence estimators and the more restrictive blind-spot ones.

### 3.5 PROPERTIES SHARED BY ALL CONSTRAINT SETS

For conciseness, $\mathcal{S}^c$ denotes one of the following sets: $\mathcal{S}_{\mathrm{BS}}^c, \mathcal{S}_{\mathrm{CED}}^c$ or $\mathcal{S}_{\mathrm{DC}}^c$. This paragraph should be read by selecting one of these three sets consistently, without mixing them. As a preliminarily observation, we notice that the constraint set $\mathcal{S}^c$ admits an affine space structure, based at $c\,\mathrm{id}$, where $\mathrm{id}$ refers to the identity map on $\mathbb{R}^n$. This statement is formalized in the following lemma (all the proofs of this paper are given in Appendix C).

**Lemma 1.** *$\mathcal{S}^0$ is a linear space and $\mathcal{S}^c$ is an affine space with $\mathcal{S}^c = c\,\mathrm{id} + \mathcal{S}^0$.*

A direct consequence (see Proposition 1) is that the optimal denoiser within each class can be written as an affine combination of the identity function and the minimizer in $\mathcal{S}^0$ of the measurement consistency term. Thus, it is sufficient to restrict the search to estimators in $\mathcal{S}^0$, since any optimal denoiser in $\mathcal{S}^c$ can be recovered from an optimal denoiser in $\mathcal{S}^0$.

**Proposition 1.** *The solution of (8) for the constraint set $\mathcal{S}^c$ can be expressed as*

$$\arg\min_{f \in \mathcal{S}^c} \mathbb{E}_{\boldsymbol{y}}\|f(\boldsymbol{y}) - \boldsymbol{y}\|_2^2 = c\,\mathrm{id} + (1-c) \arg\min_{f \in \mathcal{S}^0} \mathbb{E}_{\boldsymbol{y}}\|f(\boldsymbol{y}) - \boldsymbol{y}\|_2^2. \tag{12}$$

An immediate question that arises at this point is: *How to choose the constant $c$ to achieve the best denoising?* Proposition 2 provides a theoretical characterization of the optimal constant $c^*$.

**Proposition 2** (Optimal constant). *In the AWGN setting (see equation 2), if $\mathcal{S}^c \in \{\mathcal{S}_{\mathrm{BS}}^c, \mathcal{S}_{\mathrm{DC}}^c\}$ or if $\mathcal{S}^c = \mathcal{S}_{\mathrm{CED}}^c$ and $p_\sigma$ is a Dirac then*

$$c^* = \arg\min_{c \in \mathbb{R}} \min_{f \in \mathcal{S}^c} \mathbb{E}_{\boldsymbol{x},\boldsymbol{y}}\|f(\boldsymbol{y}) - \boldsymbol{x}\|_2^2 = 1 - \frac{n\mathbb{E}_\sigma[\sigma^2]}{\min_{f \in \mathcal{S}^0} \mathbb{E}_{\boldsymbol{y}}\|f(\boldsymbol{y}) - \boldsymbol{y}\|_2^2} \in [0, 1]. \tag{13}$$

Interestingly, the optimal constant $c^*$ lies in $[0, 1]$. As a consequence, the affine combination in Prop. 1 is in fact a convex combination in the optimal case. Naturally, $c^*$ depends on the knowledge of $\mathbb{E}_\sigma[\sigma^2]$, which may be unknown in some settings. This explains why the arbitrary choice $c = 0$ is made in practice (Batson & Royer, 2019; Tachella et al., 2025a). Unless otherwise stated, we adopt the same choice for CENSURE.

Finally, using the fact that $\mathcal{S}_{\mathrm{DC}}^c \subset \mathcal{S}_{\mathrm{CED}}^c$, we can derive a lower bound on the reconstruction error for divergence-constant estimators in the case where $\sigma$ is constant.

**Proposition 3** (A lower bound). *In the AWGN setting with constant noise level $\sigma$ (see equation 2),*

$$\min_{f \in \mathcal{S}_{\mathrm{DC}}^c} \mathbb{E}_{\boldsymbol{x},\boldsymbol{y}} \frac{1}{n} \|f(\boldsymbol{y}) - \boldsymbol{x}\|_2^2 \geq \min_{f \in \mathcal{S}_{\mathrm{CED}}^c} \mathbb{E}_{\boldsymbol{x},\boldsymbol{y}} \frac{1}{n} \|f(\boldsymbol{y}) - \boldsymbol{x}\|_2^2$$

$$= \mathrm{MMSE} + \frac{\sigma^2}{1 - \frac{\mathrm{MMSE}}{\sigma^2}} \left( c - \frac{\mathrm{MMSE}}{\sigma^2} \right)^2 \geq \mathrm{MMSE}, \tag{14}$$

*where* $\mathrm{MMSE} = \mathbb{E}_{\boldsymbol{x},\boldsymbol{y}} \frac{1}{n} \|\mathbb{E}(\boldsymbol{x}|\boldsymbol{y}) - \boldsymbol{x}\|_2^2$ *is the minimum mean square error.*

## 4 DESIGN OF DIVERGENCE-FREE NEURAL NETWORKS

We now present our proposed methodology for constructing divergence-free network architectures.

### 4.1 REPRESENTING DIVERGENCE-FREE VECTOR FIELDS

Lemma 2 offers a straightforward method for generating divergence-free vector fields and highlights the key role played by skew-symmetric matrices in ensuring zero divergence.

**Lemma 2** (A simple divergence-free vector field). *Let $\psi : \mathbb{R}^n \to \mathbb{R}$ be a smooth scalar field and let $\boldsymbol{A} \in \mathbb{R}^{n \times n}$ be a skew-symmetric matrix, i.e. $\boldsymbol{A}^\top = -\boldsymbol{A}$. The vector field $\boldsymbol{A}\nabla\psi$ is divergence-free.*

Nevertheless, this construction does not capture all divergence-free vector fields, except for the case $n = 2$. In fact, fully representing such fields typically requires combining multiple expressions of this form, as formalized in the following representer theorem.

**Theorem 1** (A universal representation of divergence-free fields). *Let $f : \mathbb{R}^n \to \mathbb{R}^n$ be a smooth vector field and let $\{\boldsymbol{A}_1, \ldots, \boldsymbol{A}_K\} \in \mathbb{R}^{n \times n}$ be a basis of the space of real skew-symmetric $n \times n$ matrices, with $n \geq 2$. $f$ is divergence-free if and only if there exist smooth scalar fields $\psi_1, \ldots, \psi_K : \mathbb{R}^n \to \mathbb{R}$ such that*

$$f = \sum_{k=1}^{K} \boldsymbol{A}_k \nabla \psi_k \,. \tag{15}$$

The proof is an extension of the work of Richter-Powell et al. (2022) which builds on the classical Hodge decomposition theorem (Morita, 2001; Berger, 2003). Note that the space of real skew-symmetric $n \times n$ matrices is of dimension $K = \binom{n}{2}$, hence the above construction uses a number of scalar fields that scales quadratically with the dimension $n$.

**Application for** $n = 3$   Consider the following basis of real skew-symmetric $3 \times 3$ matrices:

$$(\boldsymbol{A}_1, \boldsymbol{A}_2, \boldsymbol{A}_3) = \left( \begin{pmatrix} 0 & 0 & 0 \\ 0 & 0 & 1 \\ 0 & -1 & 0 \end{pmatrix}, \begin{pmatrix} 0 & 0 & -1 \\ 0 & 0 & 0 \\ 1 & 0 & 0 \end{pmatrix}, \begin{pmatrix} 0 & 1 & 0 \\ -1 & 0 & 0 \\ 0 & 0 & 0 \end{pmatrix} \right), \tag{16}$$

and let $f$ be a smooth divergence-free vector field. According to Theorem 1, there exist $\psi_1, \psi_2, \psi_3 : \mathbb{R}^3 \to \mathbb{R}$, such that

$$f = \sum_{k=1}^{3} \boldsymbol{A}_k \nabla \psi_k = \begin{pmatrix} \frac{\partial \psi_3}{\partial y_2} - \frac{\partial \psi_2}{\partial y_3} \\ \frac{\partial \psi_1}{\partial y_3} - \frac{\partial \psi_3}{\partial y_1} \\ \frac{\partial \psi_2}{\partial y_1} - \frac{\partial \psi_1}{\partial y_2} \end{pmatrix} \triangleq \nabla \times \begin{pmatrix} \psi_1 \\ \psi_2 \\ \psi_3 \end{pmatrix} \,. \tag{17}$$

In other words, there exists a vector field $\psi = (\psi_1, \psi_2, \psi_3)$ such that $f$ is its curl. This is a well-known result in the literature (Morita, 2001).

### 4.2 PROPOSED ARCHITECTURE

Our goal is to construct a parameterized function $f$, under the form of a neural network, that is divergence-free by design and whose architecture is tailored for image processing tasks, in particular denoising. To this end, we build on the representer Theorem 1, which suggests defining $f$ as a structured combination of conservative fields $\nabla\psi_k$. However, as previously noted, the number of terms

in this representation, namely $K$ in Theorem 1, is on the order of $n^2$, which becomes prohibitive as soon as we work with images. This scalability issue was previously highlighted by Richter-Powell et al. (2022). To keep computations tractable, we propose to constraint $f$ to be represented using a sparse combination of conservative fields, which we assume retains sufficient representational fidelity. This deliberate simplification ultimately consists in substituting $K$ with $K' \ll K$ (typically $K' = 8$). More precisely, we build $f$ under the form

$$f = \sum_{k=1}^{K'} \boldsymbol{A}_k \nabla \psi_k \,, \tag{18}$$

where $\psi_1, \ldots, \psi_{K'} : \mathbb{R}^n \to \mathbb{R}$ are parameterized via a single shared neural network and $\{\boldsymbol{A}_1, \ldots, \boldsymbol{A}_{K'}\} \in \mathbb{R}^{n \times n}$ are (sparse) skew-symmetric matrices, also parameterized. Note that even though we retain fewer terms in (18) than required, the resulting function remains exactly divergence-free, as ensured by Lemma 1 and 2 (divergence-free functions are closed under addition). The parameter $K'$ only affects expressivity: when $K' = 0$, $f = 0$ which is trivially divergence-free but not expressive at all; conversely, $K' = K$ yields maximal expressivity, as stated in Theorem 1, but becomes computationally intractable. We now detail the construction of both types of parameterization.

### 4.2.1 DESIGN OF THE SKEW-SYMMETRIC MATRICES

For the sake of computational efficiency, the skew-symmetric matrices $\boldsymbol{A}_k$ in (18) are chosen to be sparse matrices with shared parameters as follows:

$$\boldsymbol{A}_k = \boldsymbol{P}_k^\top \frac{\boldsymbol{\Theta} - \boldsymbol{\Theta}^\top}{2} \boldsymbol{P}_k \,, \tag{19}$$

where $\boldsymbol{\Theta}$ is a shared parameterized repeated-block diagonal matrix and where each $\boldsymbol{P}_k \in \mathbb{R}^{n \times n}$ is a different and fixed permutation matrix (typically a rotation or shift matrix). Note that the matrices $\boldsymbol{A}_k$ are guaranteed to be skew-symmetric by design thanks to the following equality of sets, valid for any permutation matrix $\boldsymbol{P}_k$: $\{\boldsymbol{A} \in \mathbb{R}^{n \times n} : \boldsymbol{A}^\top = -\boldsymbol{A}\} = \{\boldsymbol{P}_k^\top \frac{\boldsymbol{A} - \boldsymbol{A}^\top}{2} \boldsymbol{P}_k : \boldsymbol{A} \in \mathbb{R}^{n \times n}\}$.

### 4.2.2 DESIGN OF THE SCALAR FIELDS

The idea of designing neural networks to represent exact conservative fields, *i.e.* of the form $\nabla \psi$, has already been explored in works targeting energy based models or plug-and-play methods (Salimans & Ho, 2021; Hurault et al., 2022a). They all point out the critical choice of the architecture for the scalar potential function $\psi$ in order to achieve good performance in practice. In particular, as experimentally observed, modeling $\psi$ as a standard feedforward network, such as the ones used for classification, severely degrades performance. Instead, it is recommended to incorporate an architecture tailored to the target application directly into the design of $\psi$. This is why, we propose to consider parameterized scalar fields of the form

$$\psi_{\boldsymbol{\theta}, \boldsymbol{B}_k} : \boldsymbol{y} \in \mathbb{R}^n \mapsto \frac{1}{2} \left( \|\boldsymbol{B}_k \boldsymbol{y}\|_2^2 - \|\boldsymbol{B}_k \boldsymbol{y} - D_{\boldsymbol{\theta}}(\boldsymbol{y})\|_2^2 \right) \,, \tag{20}$$

where $\boldsymbol{B}_k \in \mathbb{R}^{n \times n}$ and $D_{\boldsymbol{\theta}} : \mathbb{R}^n \to \mathbb{R}^n$ is a neural network specific to image processing, typically a U-Net (Ronneberger et al., 2015). Please note that the neural network parameters $\boldsymbol{\theta}$ are shared for all scalar fields. The specific form of the scalar fields in (20) is strongly inspired by Hurault et al. (2022a), with the addition of the $\boldsymbol{B}_k$ matrices introduced in our formulation. It is justified by the fact that

$$\nabla \psi_{\boldsymbol{\theta}, \boldsymbol{B}_k}(\boldsymbol{y}) = \boldsymbol{B}_k^\top D_{\boldsymbol{\theta}}(\boldsymbol{y}) + \mathbf{J}_{D_{\boldsymbol{\theta}}}(\boldsymbol{y})^\top (\boldsymbol{B}_k \boldsymbol{y} - D_{\boldsymbol{\theta}}(\boldsymbol{y})) \,, \tag{21}$$

for which the first term is known to be effective for learning denoising functions. Beyond introducing diversity for the scalar potential functions, the inclusion of the matrix $\boldsymbol{B}_k$ in (20) has the effect of replacing the term $D_{\boldsymbol{\theta}}(\boldsymbol{y})$ by $\boldsymbol{B}_k^\top D_{\boldsymbol{\theta}}(\boldsymbol{y})$, with the hope that this (transposed) matrix could counterbalance the potentially negative effect of multiplication by a skew-symmetric matrix $\boldsymbol{A}_k$ afterwards. In practice, expression (21) is computed by differentiating (20) with respect to the input $\boldsymbol{y}$ using an automatic differentiation engine (Paszke et al., 2019), which avoids computing the full Jacobian.

Finally, the matrices $\boldsymbol{B}_k$ in (20) are parameterized analogously to (19) via a shared repeated-block diagonal matrix $\boldsymbol{\Theta}' \in \mathbb{R}^{n \times n}$, in accordance with

$$\boldsymbol{B}_k = \boldsymbol{P}_k^\top \boldsymbol{\Theta}' \boldsymbol{P}_k \,. \tag{22}$$

Table 1: The PSNR (dB) results of self-supervised methods on color datasets corrupted by synthetic white Gaussian noise. Training was conducted with **unknown non-constant** $\sigma \in [0, 75]$ (a single model to handle all noise levels). The best and second best results in each category are highlighted in red and blue colors, respectively. Non-constraint-based methods are denoted with *.

| | Dataset | Kodak24 | | | | CBSD68 | | | |
|---|---|---|---|---|---|---|---|---|---|
| | Noise level $\sigma$ | 15 | / 25 | / 50 | / 75 | 15 | / 25 | / 50 | / 75 |
| *supervised* | DRUNet *light* | 35.18 | / 32.78 | / 29.77 | / 28.14 | 34.22 | / 31.61 | / 28.42 | / 26.74 |
| *self-supervised* | Noise2Score* | 34.69 | / 32.02 | / 28.17 | / 22.91 | 33.82 | / 31.00 | / 27.16 | / 22.49 |
| | Neighbor2Neighbor* | 34.72 | / 32.37 | / 29.42 | / 27.82 | 33.82 | / 31.29 | / 28.16 | / 26.53 |
| | UNSURE ($\tau = 10^{-2}$) | 29.48 | / 22.03 | / 15.58 | / 12.56 | 29.32 | / 22.15 | / 15.71 | / 12.68 |
| | UNSURE ($\tau = 10^{-4}$) | 26.89 | / 22.01 | / 15.93 | / 12.82 | 26.83 | / 22.10 | / 16.04 | / 12.94 |
| | Noise2Self | 34.08 | / 31.90 | / 29.07 | / 27.49 | 33.06 | / 30.70 | / 27.78 | / 26.21 |
| | CENSURE (ours) | 34.21 | / 32.05 | / 29.24 | / 27.67 | 33.17 | / 30.83 | / 27.91 | / 26.33 |

Please note that the fixed permutation matrices $\boldsymbol{P}_k$ are the same as in (19). Ultimately, the learnable parameters for the proposed parametrization of (18) are $\{\boldsymbol{\theta}, \boldsymbol{\Theta}, \boldsymbol{\Theta}'\}$ and their number is only slightly greater than that of $D_{\boldsymbol{\theta}}$ since $\boldsymbol{\Theta}$ and $\boldsymbol{\Theta}'$ are sparse, which supports the practicality of our proposed parameterization.

## 5 EXPERIMENTAL RESULTS

We demonstrate the effectiveness of our proposed methodology to construct divergence-free networks, termed CENSURE (Concealed and Erratic Noise level with Stein's Unbiased Risk Estimate), in the case of self-supervised image denoising under the assumption of additive white Gaussian noise and compare its competitiveness with related state-of-the-art approaches that rely on (4) such as MC-SURE (Soltanayev & Chun, 2018) and its constraint-based variants, namely Noise2Self (Batson & Royer, 2019) and UNSURE (Tachella et al., 2025a). We also compare with Neighbor2Neighbor (Huang et al., 2021) and Noise2Score (Kim & Ye, 2021) (in the unknown–noise-level setting, its extension (Kim et al., 2022) was considered, still referred to as Noise2Score) as non-constraint-based baselines. For a fair comparison, we used a common backbone architecture for all methods and trained all models ourselves. Note that approaches that rely on a Monte Carlo approximation (6) of the divergence involve an additional hyperparameter $\tau$. In our experiments, we considered two values: $\tau = 10^{-2}$, as recommended for vectors with entries in $[0, 1]$ by Tachella et al. (2025b), and $\tau = 10^{-4}$, a value we selected based on test-set performance and which may be viewed as an oracle hyperparameter. Performance of CENSURE and other methods are assessed in terms of PSNR values. Details about training, datasets and implementations can be found in Appendix B. Source code is available at the following repository: https://github.com/sherbret/divergence_free_nn.

### 5.1 UNKNOWN NON-CONSTANT NOISE LEVEL

We first evaluate the self-supervised constraint-based methods for denoising under the assumption that the noise level $\sigma$ is unknown and varies across the training data. This scenario is the most realistic in practice because real-world sensors rarely operate at a fixed signal-to-noise ratio, and the corruption level can change over time, across devices, or even between consecutive acquisitions. According to Figure 1, our approach is, in principle, the most suitable in this setting. For each method, we learn a single model to handle all noise levels. Specifically, training is conducted using only noisy images synthetically corrupted with Gaussian noise and random $\sigma \sim \mathcal{U}([0, 75])$. Please note that the values of $\sigma$ (assumed unknown) are not used in the loss function or during inference/test time. Table 1 reports the PSNR results on two benchmark datasets for color image denoising, with representative qualitative examples shown in Figure 2. Results for grayscale images are presented in Table 3 in Appendix, along with corresponding qualitative comparisons in Figure 7. We can observe that CENSURE outperforms its constraint-based counterparts, which is in accordance with the theory. In particular, UNSURE fails when $\sigma$ varies, mainly because it tends to overfit the noise. Indeed, contrary to the case where $\sigma$ is constant, $\mathbb{E}_{\boldsymbol{y},\sigma}[\sigma^2 \operatorname{div} f(\boldsymbol{y})] \neq \mathbb{E}_{\sigma}[\sigma^2]\mathbb{E}_{\boldsymbol{y}}[\operatorname{div} f(\boldsymbol{y})]$ in full gener-

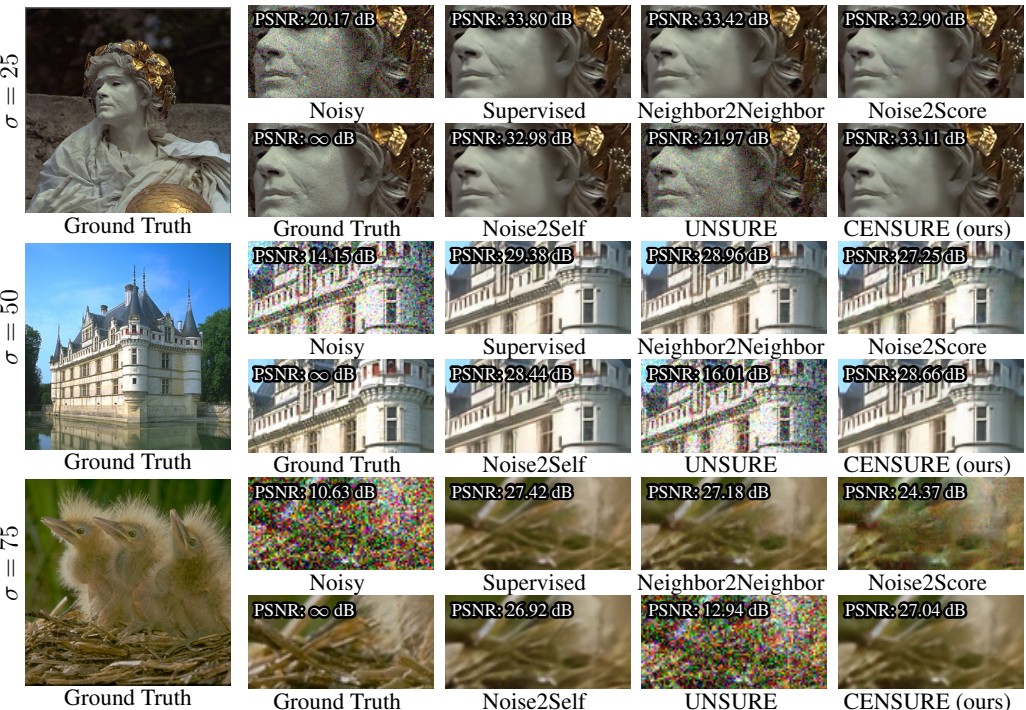

Figure 2: Qualitative blind image denoising results. All models were trained on a dataset with unknown random $\sigma \in [0, 75]$ (a single model to handle all noise levels). Best viewed by zooming.

ality since $\sigma^2$ and $\mathrm{div}\, f(\boldsymbol{y})$ are statistically dependent. Therefore, condition (7) is not satisfied for UNSURE. Finally, although Neighbor2Neighbor appears to be the best-performing self-supervised method overall, it is inherently limited to natural images, as it relies on the core assumption that two noise-free neighboring pixels share similar values most of the time. In contrast, constraint-based approaches are more general for denoising problems.

## 5.2 CONSTANT NOISE LEVEL

For the sake of completeness, we also assess the performance of self-supervised constraint-based methods in the less realistic setting where the noise level $\sigma$ is constant (whether known or unknown) across the training data. As shown in Figure 1, this scenario favors the use of estimators that are more expressive than CENSURE such as UNSURE or MC-SURE. Table 4 in Appendix D reports a quantitative comparison on two test datasets for grayscale images. Contrary to the previous scenario, a separate model is learned for each noise level. The results are organized into two categories: methods that require the noise level $\sigma$ during training or inference (known $\sigma$ setting), and those that are completely agnostic to it (unknown $\sigma$ setting), including CENSURE, UNSURE and Noise2Self. Importantly, Propositions 1 and 2 show that divergence-free denoisers, either everywhere or in expectation, can be converted into noise-level–aware ones (marked with symbol † in Table 4) without additional training. Specifically, for a divergence-free estimator $f$, the quantity $\min_{f \in \mathcal{S}^0} \mathbb{E}_{\boldsymbol{y}} \| f(\boldsymbol{y}) - \boldsymbol{y} \|_2^2$ in Proposition 2 can be approximated using a single realization $\| f(\boldsymbol{y}) - \boldsymbol{y} \|_2^2$, where $\boldsymbol{y}$ denotes the noisy input image (averaging this term over more image realizations did not lead to further improvements).

As expected, UNSURE achieves the best performance among divergence-based methods when the noise level $\sigma$ is constant and unknown, followed by CENSURE and then Noise2Self. This ordering is consistent with the fact that $\mathcal{S}^0_{\mathrm{BS}} \subset \mathcal{S}^0_{\mathrm{DC}} \subset \mathcal{S}^0_{\mathrm{CED}}$: the fewer constraints imposed on the search space of the estimator, the more expressive it becomes, as shown by Figure 1. Note that a slight performance gap is observed for UNSURE depending on the choice of the hyperparameter $\tau$.

When the noise level $\sigma$ is assumed known, all divergence-free methods naturally benefit from this additional information, showing PSNR gains in line with theoretical expectations (see Subsection 3.5), except for UNSURE with $\tau = 10^{-2}$. This hyperparameter choice is also suboptimal for MC-SURE,

whose performance falls short of expectations, allowing CENSURE to outperform it in most cases. Indeed, according to (4), a denoiser trained with the SURE loss should achieve performance comparable to its supervised counterpart. The observed underperformance can largely be attributed to stability issues during training. As illustrated in Figure 4 in Appendix, the training curves of MC-SURE and UNSURE exhibit pronounced fluctuations, in contrast to the much smoother trajectories of the other methods, including ours. This phenomenon was already described in Soltanayev & Chun (2018) and considering the oracle hyperparameter $\tau = 10^{-4}$ (chosen based on test-set performance) instead solved this issue in our case. To the best of our knowledge, there is no systematic way to set this value; it likely depends on the Lipschitz constant of the network, which is itself difficult to estimate in advance Pintore & Després (2024). This highlights the strength of our approach, as CENSURE remains completely independent of this choice by naturally enforcing strict zero divergence. Finally, when the oracle value $\tau = 10^{-4}$ is used, MC-SURE becomes the best-performing method, achieving results close to the supervised model, in agreement with theoretical analysis. Qualitative examples illustrating the performance of self-supervised methods in the constant-noise regime are presented in Figure 8 in Appendix.

## 6 CONCLUSION

We presented an original approach for constraining neural networks to be divergence-free by design. Our proposed parameterization is grounded in a representer theorem for divergence-free vector fields, which characterizes them as structured combinations of conservative fields. Leveraging this theoretical foundation and incorporating sparsity constraints, we derived parameterizations for neural networks that are both resource-efficient and scalable to high-dimensional settings. The practical relevance of our approach is illustrated in the context of self-supervised image denoising, where we demonstrated that that these models achieve competitive performance compared to existing constraint-based methods, especially when the noise level is unknown and varies across the training data. Beyond denoising, our results suggest that our divergence-free parameterization may hold promise for a wider range of high-dimensional learning tasks, in particular in physics-informed machine learning, opening new avenues for future research.

### ACKNOWLEDGEMENTS

The authors acknowledge the support of the Research Group IASIS of the CNRS Informatics (Project DIVIN). Etienne Meunier was partially supported by the Research Programme "PPR Ocean & Climate" through a postdoctoral scholarship. This work was granted access to the HPC resources of IDRIS under the allocation 2025-AD011015932R1 made by GENCI. We thank Cédric Herzet for fruitful discussions.

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

# A LIMITATIONS

We would like to mention that the proposed parameterization for enforcing zero divergence entails higher computational cost during both training and inference compared to its supervised and constraint-based self-supervised counterparts (Tachella et al., 2025a; Soltanayev & Chun, 2018; Batson & Royer, 2019). In particular, our method requires one feedforward pass through the backbone network along with $K'$ gradient computations at inference ($K' = 8$ in our implementation). Since each backpropagation has a cost comparable to a forward pass (Hurault et al., 2022a), the overall inference cost amounts to roughly $K' + 1$ times that of a single feedforward evaluation, which may limit its applicability in time-sensitive settings. Moreover, due to scalability constraints, we deliberately restricted the number of terms in the sum (15) from Theorem 1. While this sparsity constraint may not fully capture the underlying optimal solution, we demonstrated that it still yields strong performance in image denoising. Finally, our application in image denoising targets only additive white Gaussian noise, and its effectiveness under alternative noise models remains unexplored. We nevertheless demonstrate in Figure 9 that the Gaussian denoiser learned with our approach can be directly applied to real-world noisy fluorescence images, when combined with a variance-stabilizing transformation learned using the Noise2VST (Herbreteau & Unser, 2025) framework. Extending our parameterization to handle other types of noise, such as Poisson–Gaussian corruption, presents a promising direction for future research.

Table 2: Execution time (in seconds) comparison on a batch of size $16 \times 1 \times 64 \times 64$ for different approaches with shared backbone architecture (GPU: Tesla V100).

| Model | Training ↓ | Inference ↓ | # parameters |
|---|---|---|---|
| Supervised | 0.036 | 0.007 | $17,007,744$ |
| MC-SURE | 0.061 | 0.007 | $17,007,744$ |
| UNSURE | 0.062 | 0.007 | $17,007,744$ |
| Noise2Self | 0.036 | 0.120 | $17,007,744$ |
| CENSURE $K' = 2$ | 0.356 | 0.024 | $17,008,256$ |
| CENSURE $K' = 4$ | 0.679 | 0.039 | $17,008,256$ |
| CENSURE $K' = 6$ | 1.002 | 0.055 | $17,008,256$ |
| CENSURE $K' = 8$ | 1.323 | 0.072 | $17,008,256$ |

# B IMPLEMENTATION DETAILS

**Common backbone architecture** For a fair comparison, we adopt a variant of the DRUNet architecture (Zhang et al., 2022) as the shared backbone across all approaches. In the original formulation, each scale is composed of four residual blocks of the form "$3 \times 3$ conv $\rightarrow$ ReLU $\rightarrow 3 \times 3$ conv". To reduce computational cost, we limit this to two residual blocks per scale. Moreover, as in Hurault et al. (2022b), we replace ReLU by Softplus activations with sharpness parameter $\beta = 100$, which acts as a smooth surrogate for ReLU, easing training for conservative field networks (Hurault et al., 2022a). Note that the noise level map was removed from the original architecture.

**Datasets** All models were trained using the same large-scale dataset proposed in Zhang et al. (2022), which contains a total of 8,694 images. This includes 400 images from the Berkeley Segmentation Dataset (BSD400) (Martin et al., 2001), 4,744 images from the Waterloo Exploration Database (Ma et al., 2017), 900 images from DIV2K (Agustsson & Timofte, 2017), and 2,750 images from Flickr2K (Lim et al., 2017). The training set is augmented through random horizontal and vertical flips as well as random rotations of $90°$. For validation, we use the BSD32 dataset (Martin et al., 2001), consisting of 32 images, to monitor training progress and select the best-performing model. Finally, the evaluation for color images is carried out on two test sets, Kodak24 and BSD68 (Martin et al., 2001), which are completely separate from both the training and validation data. For grayscale images, the test set Kodak24 is replaced by Set12.

**Training details** All models are trained for 600,000 iterations, where each training iteration involves a gradient-based pass on a batch of patches of size $128 \times 128$ that are randomly cropped from

training images (except for CENSURE where patches are taken of size $64 \times 64$ in order to accelerate training). We use a batch size of 16 and employ the Adam optimizer (Kingma & Ba, 2015) with an initial learning rate of $10^{-4}$ as in Zhang et al. (2022), which is halved every 150,000 iterations. Approaches that rely on a Monte Carlo approximation (6) of the divergence involve an additional hyperparameter $\tau$. In our experiments, we considered two values: $\tau = 10^{-2}$, as recommended for vectors with entries in $[0, 1]$ by Tachella et al. (2025b), and $\tau = 10^{-4}$, a value we selected based on test-set performance and which may be viewed as an oracle hyperparameter. For the UNSURE (Tachella et al., 2025a), Noise2Score (Kim & Ye, 2021), and Neighbor2Neighbor (Huang et al., 2021) losses, we followed the default settings of the DeepInverse library (Tachella et al., 2025b); in particular, the momentum parameter for the gradient ascent on the noise level was fixed at 0.9 for UNSURE and the minimum and maximum noise level for Noise2Score, which are annealed during training, are set to 0.001 and 0.1, respectively.

**Implementation choices for CENSURE** Our proposed parameterization (18) of divergence-free estimators requires several additional hyperparameters that must be specified, including the number of terms $K'$ in the sum, the size $\kappa \times \kappa$ of the blocks in the two repeated-block diagonal matrices $\boldsymbol{\Theta}$ and $\boldsymbol{\Theta}'$, and the selection of the permutation matrices $\boldsymbol{P}_k$. First of all, in order to drastically reduce the computational burden, we set $K' = 8$. The block size $\kappa$ is chosen as $4 \times 4 \times C$, where $C$ is the number of channels ($C = 1$ for grayscale images and $C = 3$ for color ones), resulting in a total of $512 \times C$ additional learnable parameters, which is negligible compared to the 17M parameters of the network backbone. Finally, the first four permutation matrices $\boldsymbol{P}_k$ were selected to perform circular horizontal shifts of the input image by 0 to 3 pixels, while the remaining four are obtained by composing these shifts with a $90°$ rotation.

Our implementation is written in Python using the PyTorch framework (Paszke et al., 2019) and with additional support from the DeepInverse library (Tachella et al., 2025b). Training was conducted on a Tesla V100 GPU.

## C  PROOFS

*Proof of Lemma 1.* $\mathcal{S}^0$ is a linear space due to the linearity of the partial derivative operator and the linearity of expectation. Let $f \in c\, \mathrm{id}_n + \mathcal{S}^0$. For all $\boldsymbol{y} \in \mathbb{R}^n$, $\frac{\partial [c\, \mathrm{id}_n]_i}{\partial y_i}(\boldsymbol{y}) = c$ and so $\mathrm{div}(c\, \mathrm{id}_n)(\boldsymbol{y}) = nc$. Therefore, $f \in \mathcal{S}^c$. Reciprocally, let $f \in \mathcal{S}^c$. Then, $f = c\, \mathrm{id}_n + (f - c\, \mathrm{id}_n) \in c\, \mathrm{id}_n + \mathcal{S}^0$ by linearity of the partial derivative operator and the linearity of expectation. $\square$

*Proof of Proposition 1.* According to Lemma 1, $\mathcal{S}^c = c\, \mathrm{id} + \mathcal{S}^0$. In particular, if $c \neq 1$, $\mathcal{S}^c = c\, \mathrm{id} + (1-c)\mathcal{S}^0$ and we have

$$\underset{f \in \mathcal{S}^c}{\arg\min} \, \mathbb{E}_{\boldsymbol{y}} \|f(\boldsymbol{y}) - \boldsymbol{y}\|_2^2 = \underset{f \in c\, \mathrm{id} + (1-c)\mathcal{S}^0}{\arg\min} \, \mathbb{E}_{\boldsymbol{y}} \|f(\boldsymbol{y}) - \boldsymbol{y}\|_2^2 \tag{23}$$

$$= c\, \mathrm{id} + (1-c) \underset{f \in \mathcal{S}^0}{\arg\min} \, \mathbb{E}_{\boldsymbol{y}} \|c\, \mathrm{id}(\boldsymbol{y}) + (1-c)f(\boldsymbol{y}) - \boldsymbol{y}\|_2^2 \tag{24}$$

$$= c\, \mathrm{id} + (1-c) \underset{f \in \mathcal{S}^0}{\arg\min} \, (1-c)^2 \mathbb{E}_{\boldsymbol{y}} \|f(\boldsymbol{y}) - \boldsymbol{y}\|_2^2 \tag{25}$$

$$= c\, \mathrm{id} + (1-c) \underset{f \in \mathcal{S}^0}{\arg\min} \, \mathbb{E}_{\boldsymbol{y}} \|f(\boldsymbol{y}) - \boldsymbol{y}\|_2^2 . \tag{26}$$

This still holds for $c = 1$ since we have trivially $\underset{f \in \mathcal{S}^1}{\arg\min} \, \mathbb{E}_{\boldsymbol{y}} \|f(\boldsymbol{y}) - \boldsymbol{y}\|_2^2 = \mathrm{id}$. $\square$

*Proof of Proposition 2.* According to (4) (Stein, 1981),

$$\mathbb{E}_{\boldsymbol{x},\boldsymbol{y}} \|f(\boldsymbol{y}) - \boldsymbol{x}\|_2^2 = \mathbb{E}_{\boldsymbol{y},\sigma} \left[ -n\sigma^2 + \|f(\boldsymbol{y}) - \boldsymbol{y}\|_2^2 + 2\sigma^2 \, \mathrm{div}\, f(\boldsymbol{y}) \right] . \tag{27}$$

Moreover, if $\mathcal{S}^c \in \{\mathcal{S}_{\mathrm{BS}}^c, \mathcal{S}_{\mathrm{DC}}^c\}$ or if $\mathcal{S}^c = \mathcal{S}_{\mathrm{CED}}^c$ and $p_\sigma$ is a Dirac then

$$\forall f \in \mathcal{S}^c, \quad \mathbb{E}_{\boldsymbol{y},\sigma} \left[ \sigma^2 \, \mathrm{div}\, f(\boldsymbol{y}) \right] = nc \mathbb{E}_\sigma [\sigma^2] , \tag{28}$$

hence,

$$\underset{f \in \mathcal{S}^c}{\min} \, \mathbb{E}_{\boldsymbol{x},\boldsymbol{y}} \|f(\boldsymbol{y}) - \boldsymbol{x}\|_2^2 = -n\mathbb{E}_\sigma [\sigma^2] + 2nc \mathbb{E}_\sigma [\sigma^2] + \underset{f \in \mathcal{S}^c}{\min} \, \mathbb{E}_{\boldsymbol{y}} \|f(\boldsymbol{y}) - \boldsymbol{y}\|_2^2 . \tag{29}$$

According to Lemma 1, $\mathcal{S}^c = c\,\mathrm{id} + \mathcal{S}^0$. In particular, if $c \neq 1$, $\mathcal{S}^c = c\,\mathrm{id} + (1-c)\mathcal{S}^0$ and we have

$$\min_{f \in \mathcal{S}^c} \mathbb{E}_{\boldsymbol{y}} \|f(\boldsymbol{y}) - \boldsymbol{y}\|_2^2 = \min_{f \in c\,\mathrm{id} + (1-c)\mathcal{S}^0} \mathbb{E}_{\boldsymbol{y}} \|f(\boldsymbol{y}) - \boldsymbol{y}\|_2^2 \tag{30}$$

$$= \min_{f \in \mathcal{S}^0} \mathbb{E}_{\boldsymbol{y}} \|c\boldsymbol{y} + (1-c)f(\boldsymbol{y}) - \boldsymbol{y}\|_2^2 \tag{31}$$

$$= (1-c)^2 \min_{f \in \mathcal{S}^0} \mathbb{E}_{\boldsymbol{y}} \|f(\boldsymbol{y}) - \boldsymbol{y}\|_2^2 . \tag{32}$$

This still holds for $c = 1$ as $\min_{f \in \mathcal{S}^1} \mathbb{E}_{\boldsymbol{y}} \|f(\boldsymbol{y}) - \boldsymbol{y}\|_2^2 = 0$ since $\mathrm{id} \in \mathcal{S}^1$.

Finally, for all $c \in \mathbb{R}$,

$$\min_{f \in \mathcal{S}^c} \mathbb{E}_{\boldsymbol{x},\boldsymbol{y}} \|f(\boldsymbol{y}) - \boldsymbol{x}\|_2^2 = -n\mathbb{E}_{\sigma}[\sigma^2] + 2nc\mathbb{E}_{\sigma}[\sigma^2] + (1-c)^2 \min_{f \in \mathcal{S}^0} \mathbb{E}_{\boldsymbol{y}} \|f(\boldsymbol{y}) - \boldsymbol{y}\|_2^2 . \tag{33}$$

As a consequence,

$$\arg\min_{c \in \mathbb{R}} \min_{f \in \mathcal{S}^c} \mathbb{E}_{\boldsymbol{x},\boldsymbol{y}} \|f(\boldsymbol{y}) - \boldsymbol{x}\|_2^2 = 1 - \frac{n\mathbb{E}_{\sigma}[\sigma^2]}{\min_{f \in \mathcal{S}^0} \mathbb{E}_{\boldsymbol{y}} \|f(\boldsymbol{y}) - \boldsymbol{y}\|_2^2} . \tag{34}$$

But this latter quantity lies in $[0, 1]$ since, in particular for $c = 0$ in (29),

$$\min_{f \in \mathcal{S}^0} \mathbb{E}_{\boldsymbol{y}} \|f(\boldsymbol{y}) - \boldsymbol{y}\|_2^2 = n\mathbb{E}_{\sigma}[\sigma^2] + \min_{f \in \mathcal{S}^0} \mathbb{E}_{\boldsymbol{x},\boldsymbol{y}} \|f(\boldsymbol{y}) - \boldsymbol{x}\|_2^2 > 0 , \tag{35}$$

and so,

$$\frac{n\mathbb{E}_{\sigma}[\sigma^2]}{\min_{f \in \mathcal{S}^0} \mathbb{E}_{\boldsymbol{y}} \|f(\boldsymbol{y}) - \boldsymbol{y}\|_2^2} = \frac{n\mathbb{E}_{\sigma}[\sigma^2]}{n\mathbb{E}_{\sigma}[\sigma^2] + \min_{f \in \mathcal{S}^0} \mathbb{E}_{\boldsymbol{x},\boldsymbol{y}} \|f(\boldsymbol{y}) - \boldsymbol{x}\|_2^2} \in [0, 1] . \tag{36}$$

$\square$

*Proof of Proposition 3.* According to (29), for all $c \in \mathbb{R}$,

$$\min_{f \in \mathcal{S}_{\mathrm{CED}}^c} \mathbb{E}_{\boldsymbol{x},\boldsymbol{y}} \tfrac{1}{n} \|f(\boldsymbol{y}) - \boldsymbol{x}\|_2^2 = -\sigma^2 + 2\sigma^2 c + (1-c)^2 \min_{f \in \mathcal{S}_{\mathrm{CED}}^0} \mathbb{E}_{\boldsymbol{y}} \tfrac{1}{n} \|f(\boldsymbol{y}) - \boldsymbol{y}\|_2^2 . \tag{37}$$

But since $f^{\mathrm{ZED}} = \arg\min_{f \in \mathcal{S}_{\mathrm{CED}}^0} \mathbb{E}_{\boldsymbol{y}} \tfrac{1}{n} \|f(\boldsymbol{y}) - \boldsymbol{y}\|_2^2$ has a closed-from expression (Tachella et al., 2025a), namely $f^{\mathrm{ZED}}(\boldsymbol{y}) = \boldsymbol{y} + \hat{\eta} \nabla \log p_{\boldsymbol{y}}(\boldsymbol{y})$, with $\hat{\eta} = (\mathbb{E}_{\boldsymbol{y}} \tfrac{1}{n} \|\nabla \log p_{\boldsymbol{y}}(\boldsymbol{y})\|_2^2)^{-1}$, we have

$$\min_{f \in \mathcal{S}_{\mathrm{CED}}^0} \mathbb{E}_{\boldsymbol{y}} \tfrac{1}{n} \|f(\boldsymbol{y}) - \boldsymbol{y}\|_2^2 = \mathbb{E}_{\boldsymbol{y}} \tfrac{1}{n} \|f^{\mathrm{ZED}}(\boldsymbol{y}) - \boldsymbol{y}\|_2^2 = \mathbb{E}_{\boldsymbol{y}} \tfrac{1}{n} \|\hat{\eta} \nabla \log p_{\boldsymbol{y}}(\boldsymbol{y})\|_2^2 = \hat{\eta} . \tag{38}$$

Moreover, $\hat{\eta} = \frac{\sigma^2}{1 - \frac{\mathrm{MMSE}}{\sigma^2}} \geq \sigma^2$ (Tachella et al., 2025a). Finally, since $\mathcal{S}_{\mathrm{DC}}^c \subseteq \mathcal{S}_{\mathrm{CED}}^c$,

$$\min_{f \in \mathcal{S}_{\mathrm{DC}}^c} \mathbb{E}_{\boldsymbol{x},\boldsymbol{y}} \tfrac{1}{n} \|f(\boldsymbol{y}) - \boldsymbol{x}\|_2^2 \geq \min_{f \in \mathcal{S}_{\mathrm{CED}}^c} \mathbb{E}_{\boldsymbol{x},\boldsymbol{y}} \tfrac{1}{n} \|f(\boldsymbol{y}) - \boldsymbol{x}\|_2^2 \tag{39}$$

$$= -\sigma^2 + 2\sigma^2 c + (1-c)^2 \frac{\sigma^2}{1 - \frac{\mathrm{MMSE}}{\sigma^2}} \tag{40}$$

$$= \frac{\sigma^2}{1 - \frac{\mathrm{MMSE}}{\sigma^2}} \left( c^2 - 2c + 1 + (-\sigma^2 + 2\sigma^2 c) \frac{1 - \frac{\mathrm{MMSE}}{\sigma^2}}{\sigma^2} \right) \tag{41}$$

$$= \frac{\sigma^2}{1 - \frac{\mathrm{MMSE}}{\sigma^2}} \left( c^2 - 2c + 1 + (2c - 1)\left(1 - \frac{\mathrm{MMSE}}{\sigma^2}\right) \right) \tag{42}$$

$$= \frac{\sigma^2}{1 - \frac{\mathrm{MMSE}}{\sigma^2}} \left( \left(c - \frac{\mathrm{MMSE}}{\sigma^2}\right)^2 + \frac{\mathrm{MMSE}}{\sigma^2}\left(1 - \frac{\mathrm{MMSE}}{\sigma^2}\right) \right) \tag{43}$$

$$= \mathrm{MMSE} + \frac{\sigma^2}{1 - \frac{\mathrm{MMSE}}{\sigma^2}} \left( c - \frac{\mathrm{MMSE}}{\sigma^2} \right)^2 \geq \mathrm{MMSE} . \tag{44}$$

$\square$

*Proof of Lemma 2.* Let $f : \boldsymbol{x} \mapsto \boldsymbol{A}\nabla\psi(\boldsymbol{x})$. We want to compute $\operatorname{div} f(\boldsymbol{x}) = \operatorname{tr}(\nabla f(\boldsymbol{x}))$ for all $\boldsymbol{x} \in \mathbb{R}^n$. We have $f = \varphi_2 \circ \varphi_1$, with for all $\boldsymbol{x} \in \mathbb{R}^n$,

| $\varphi_1(\boldsymbol{x}) = \nabla\psi(\boldsymbol{x})$ | $\nabla\varphi_1(\boldsymbol{x}) = \boldsymbol{H}_\psi(\boldsymbol{x})$ |
|---|---|
| $\varphi_2(\boldsymbol{x}) = \boldsymbol{A}\boldsymbol{x}$ | $\nabla\varphi_2(\boldsymbol{x}) = \boldsymbol{A}^\top$ |

where $\boldsymbol{H}_\psi(\boldsymbol{x})$ denotes the Hessian matrix of $\psi$ evaluated at $\boldsymbol{x}$. According to the chain rule (Bertsekas, 1995),

$$\nabla f(\boldsymbol{x}) = \nabla\varphi_1(\boldsymbol{x})\nabla\varphi_2(\varphi_1(\boldsymbol{x})) = \boldsymbol{H}_\psi(\boldsymbol{x})\boldsymbol{A}^\top , \tag{45}$$

hence $\operatorname{div} f(\boldsymbol{x}) = \operatorname{tr}(\boldsymbol{H}_\psi(\boldsymbol{x})\boldsymbol{A}^\top) = -\operatorname{tr}(\boldsymbol{A}\boldsymbol{H}_\psi(\boldsymbol{x})) = 0$. Indeed, the trace of the product of a skew-symmetric and a symmetric matrix is zero:

$$\operatorname{tr}(\boldsymbol{A}\boldsymbol{B}) = \operatorname{tr}((\boldsymbol{A}\boldsymbol{B})^\top) = \operatorname{tr}(\boldsymbol{B}^\top\boldsymbol{A}^\top) = -\operatorname{tr}(\boldsymbol{B}\boldsymbol{A}) = -\operatorname{tr}(\boldsymbol{A}\boldsymbol{B}). \tag{46}$$

where $\boldsymbol{B} \in \mathbb{R}^{n \times n}$ denotes a symmetric matrix. $\square$

*Proof of Theorem 1.* Assume that there exist smooth scalar fields $\psi_1, \ldots, \psi_K : \mathbb{R}^n \to \mathbb{R}$ such that $f = \sum_{k=1}^K \boldsymbol{A}_k \nabla\psi_k$. According to Lemma 2, we have that each $\boldsymbol{A}_k \nabla\psi_k$ is divergence-free. Finally, since the set of divergence-free functions is a linear space from Lemma 1, $f$ is divergence-free.

Reciprocally, assume that $f : \mathbb{R}^n \to \mathbb{R}^n$ is a smooth divergence-free vector field with $n \geq 2$. According to Richter-Powell et al. (2022) (a proof is provided below which does not involve the formalism of differential forms), there exists a smooth skew-symmetric matrix field[1] $F : \mathbb{R}^n \to \mathbb{R}^{n \times n}$ such that

$$\forall i \in \{1, \ldots n\}, \quad f_i = \sum_{j=1}^n \frac{\partial F_{i,j}}{\partial x_j} . \tag{47}$$

By denoting $\boldsymbol{E}^{(i,j)}$ the standard basis matrix of $\mathbb{R}^{n \times n}$, having a 1 in the $(i,j)$-th entry and zeros elsewhere, we can rewrite it as

$$f = \sum_{i,j} \boldsymbol{E}^{(i,j)}\nabla F_{i,j} = \sum_{i<j} \boldsymbol{E}^{(i,j)}\nabla F_{i,j} + \sum_{j<i} \boldsymbol{E}^{(i,j)}\nabla F_{i,j} + \sum_{i} \boldsymbol{E}^{(i,i)}\nabla F_{i,i} \tag{48}$$

$$= \sum_{i<j} \boldsymbol{E}^{(i,j)}\nabla F_{i,j} + \sum_{i<j} \boldsymbol{E}^{(j,i)}\nabla F_{j,i} \quad (\text{since } F_{i,i} = 0) \tag{49}$$

$$= \sum_{i<j} (\boldsymbol{E}^{(i,j)} - \boldsymbol{E}^{(i,j)\top})\nabla F_{i,j} \quad (\text{since } F_{j,i} = -F_{i,j}) \tag{50}$$

$$= \sum_{k=1}^K (\boldsymbol{E}^{\varphi(k)} - \boldsymbol{E}^{\varphi(k)\top})\nabla F_{\varphi(k)} \tag{51}$$

$$= \sum_{k=1}^K \boldsymbol{B}_k \nabla F_{\varphi(k)} , \tag{52}$$

where $\varphi$ is a bijection from $\{1, \ldots, \binom{n}{2}\}$ to $\{(i,j) \in \{1, \ldots, n\}^2 | i < j\}$ and $\boldsymbol{B}_k \triangleq \boldsymbol{E}^{\varphi(k)} - \boldsymbol{E}^{\varphi(k)\top}$.

We can notice that $(\boldsymbol{B}_1, \ldots, \boldsymbol{B}_K) \in \mathbb{R}^{n \times n}$ is nothing else than the canonical basis of the space of real skew-symmetric $n \times n$ matrices. Therefore, $\forall 1 \leq k \leq K, \exists \lambda_1^{(k)}, \ldots, \lambda_K^{(k)}$,

$$\boldsymbol{B}_k = \sum_{i=1}^K \lambda_i^{(k)} \boldsymbol{A}_i . \tag{53}$$

Hence,

$$f = \sum_{k=1}^K \left( \sum_{i=1}^K \lambda_i^{(k)} \boldsymbol{A}_i \right) \nabla F_{\varphi(k)} = \sum_{i=1}^K \boldsymbol{A}_i \nabla \left( \sum_{k=1}^K \lambda_i^{(k)} F_{\varphi(k)} \right) . \tag{54}$$

We conclude by setting $\psi_i = \sum_{k=1}^K \lambda_i^{(k)} F_{\varphi(k)}$. $\square$

---

[1] *i.e.* a map $F : \mathbb{R}^n \to \mathbb{R}^{n \times n}$ such that $\forall \boldsymbol{x} \in \mathbb{R}^n, F(\boldsymbol{x})^\top = -F(\boldsymbol{x})$.

*Proof of (47).* Let $f : \mathbb{R}^n \to \mathbb{R}^n$ be a smooth divergence-free vector field with $n \geq 2$.

Define, for all $i \in \{2, \dots, n\}$,

$$\psi_i : \boldsymbol{x} \in \mathbb{R}^n \mapsto - \int_0^{x_1} f_i(\xi, x_2, x_3, \dots, x_n) d\xi + \delta_{i2} \int_0^{x_2} f_1(0, \xi, x_3, \dots, x_n) d\xi, \quad (55)$$

where $\delta_{ij}$ denotes the Kronecker delta. By construction, $f_i = -\dfrac{\partial \psi_i}{\partial x_1}$ for all $i \in \{2, \dots, n\}$.

Moreover, for all $\boldsymbol{x} \in \mathbb{R}^n$,

$$\sum_{i=2}^n \frac{\partial}{\partial x_i} \psi_i(\boldsymbol{x}) = \frac{\partial}{\partial x_2} \int_0^{x_2} f_1(0, \xi, x_3, \dots, x_n) d\xi - \sum_{i=2}^n \frac{\partial}{\partial x_i} \int_0^{x_1} f_i(\xi, x_2, x_3, \dots, x_n) d\xi \quad (56)$$

$$= f_1(0, x_2, \dots, x_n) + \int_0^{x_1} \sum_{i=2}^n -\frac{\partial}{\partial x_i} f_i(\xi, x_2, x_3, \dots, x_n) d\xi \quad (57)$$

$$= f_1(0, x_2, \dots, x_n) + \int_0^{x_1} \frac{\partial}{\partial x_1} f_1(\xi, x_2, x_3, \dots, x_n) d\xi \quad (58)$$

$$= f_1(0, x_2, \dots, x_n) + f_1(\boldsymbol{x}) - f_1(0, x_2, \dots, x_n) \quad (59)$$

$$= f_1(\boldsymbol{x}), \quad (60)$$

where, in (58), we used the fact that $f$ is divergence-free.

Define

$$F : \boldsymbol{x} \in \mathbb{R}^n \mapsto \begin{pmatrix} 0 & \psi_2(\boldsymbol{x}) & \psi_3(\boldsymbol{x}) & \cdots & \psi_n(\boldsymbol{x}) \\ -\psi_2(\boldsymbol{x}) & 0 & 0 & \cdots & 0 \\ -\psi_3(\boldsymbol{x}) & 0 & 0 & \cdots & 0 \\ \vdots & \vdots & \vdots & \ddots & \vdots \\ -\psi_n(\boldsymbol{x}) & 0 & 0 & \cdots & 0 \end{pmatrix}. \quad (61)$$

Finally, $F$ is a smooth skew-symmetric matrix field such that

$$\forall i \in \{1, \dots n\}, \quad f_i = \sum_{j=1}^n \frac{\partial F_{i,j}}{\partial x_j}. \quad (62)$$

$\square$

# D  ADDITIONAL QUANTITATIVE RESULTS FOR GRAYSCALE IMAGES

Table 3: The PSNR (dB) results of self-supervised methods on **grayscale** datasets corrupted by synthetic white Gaussian noise. Training was conducted with **unknown non-constant** $\sigma \in [0, 50]$ (a single model to handle all noise levels). The best and second best results in each category are highlighted in red and blue colors, respectively. Non-constraint-based methods are denoted with *.

| | Dataset | Set12 | | | BSD68 | | |
|---|---|---|---|---|---|---|---|
| | Noise level $\sigma$ | 15 / | 25 / | 50 | 15 / | 25 / | 50 |
| *supervised* | DRUNet *light* | 33.13 / | 30.84 / | 27.76 | 31.84 / | 29.41 / | 26.49 |
| *self-supervised* | Noise2Score* | 32.70 / | 30.27 / | 26.37 | 31.43 / | 28.94 / | 25.43 |
| | Neighbor2Neighbor* | 32.29 / | 30.38 / | 27.44 | 31.11 / | 29.07 / | 26.30 |
| | UNSURE ($\tau = 10^{-2}$) | 26.32 / | 20.81 / | 14.99 | 26.25 / | 20.91 / | 15.14 |
| | UNSURE ($\tau = 10^{-4}$) | 23.10 / | 18.63 / | 13.11 | 23.08 / | 18.70 / | 13.21 |
| | Noise2Self | 31.05 / | 29.47 / | 26.94 | 29.24 / | 27.79 / | 25.69 |
| | CENSURE (ours) | 31.62 / | 29.80 / | 27.05 | 29.73 / | 28.13 / | 25.81 |

Table 4: The PSNR (dB) results of deep learning-based methods applied to **grayscale** datasets corrupted by synthetic white Gaussian noise. Training was conducted with **constant** noise level $\sigma$ (one model per noise level). The best and second best results in each category are highlighted in red and blue colors, respectively. Non-constraint-based methods are denoted with *. Models converted into noise-level–aware variants, using Propositions 1 and 2, are indicated by the symbol † (no additional training).

| | | Dataset | Set12 | | | BSD68 | | |
|---|---|---|---|---|---|---|---|---|
| | | Noise level $\sigma$ | 15 / | 25 / | 50 | 15 / | 25 / | 50 |
| | *supervised* | DRUNet *light* | 33.24 / | 30.92 / | 27.84 | 31.91 / | 29.46 / | 26.55 |
| *self-supervised* | **Unknown constant** $\sigma$ | Noise2Score* | 32.93 / | 30.58 / | 27.40 | 31.61 / | 29.21 / | 26.29 |
| | | Neighbor2Neighbor* | 32.84 / | 30.62 / | 27.59 | 31.63 / | 29.26 / | 26.40 |
| | | UNSURE ($\tau = 10^{-2}$) | 31.88 / | 29.84 / | 27.15 | 30.90 / | 28.72 / | 26.08 |
| | | Noise2Self | 31.15 / | 29.55 / | 27.02 | 29.29 / | 27.83 / | 25.73 |
| | | CENSURE (ours) | 31.65 / | 29.81 / | 27.05 | 29.87 / | 28.14 / | 25.78 |
| | | UNSURE ($\tau$ oracle) | 32.16 / | 30.28 / | 27.33 | 31.14 / | 28.99 / | 26.19 |
| | **Known constant** $\sigma$ | Noise2Score* | 33.04 / | 30.65 / | 27.41 | 31.80 / | 29.33 / | 26.34 |
| | | MC-SURE ($\tau = 10^{-2}$) | 32.13 / | 29.97 / | 27.27 | 31.20 / | 28.86 / | 26.22 |
| | | UNSURE† ($\tau = 10^{-2}$) | 31.80 / | 29.49 / | 27.14 | 30.58 / | 28.39 / | 26.01 |
| | | Noise2Self† | 32.07 / | 30.05 / | 27.25 | 30.80 / | 28.66 / | 26.09 |
| | | CENSURE† (ours) | 32.46 / | 30.28 / | 27.27 | 31.20 / | 28.91 / | 26.13 |
| | | MC-SURE ($\tau$ oracle) | 33.22 / | 30.87 / | 27.77 | 31.90 / | 29.44 / | 26.51 |
| | | UNSURE† ($\tau$ oracle) | 33.05 / | 30.79 / | 27.58 | 31.47 / | 29.22 / | 26.34 |

# E  TOY EXAMPLES

To gain some intuition about optimal divergence-constant estimators for denoising in $\mathcal{S}_{\mathrm{DC}}^c$ and their difference with their counterparts in $\mathcal{S}_{\mathrm{BS}}^c$ and $\mathcal{S}_{\mathrm{CED}}^c$, we extend the toy examples from Tachella et al. (2025a) to the two-dimensional case[2]. In all experiments, the noise level $\sigma$ is assumed constant.

## E.1  TWO DELTAS

Let $p_{\boldsymbol{x}} = \frac{1}{2}\delta_{\boldsymbol{1}} + \frac{1}{2}\delta_{-\boldsymbol{1}}$ denote the distribution of the clean signal $\boldsymbol{x}$ and $p_{\boldsymbol{y}} = \frac{1}{2}\mathcal{N}(-\boldsymbol{1}, \sigma^2\boldsymbol{I}) + \frac{1}{2}\mathcal{N}(\boldsymbol{1}, \sigma^2\boldsymbol{I})$ be the distribution of the signal corrupted by AWGN (see equation 2), where $\boldsymbol{1}$ is the all-ones vector in $\mathbb{R}^2$ and $\boldsymbol{I}$ denotes the $2 \times 2$ identity matrix. Since the prior distribution is explicit, we can compute closed-form solutions for the following estimators:

$$\begin{cases} f^{\mathrm{MMSE}} = \arg\min_f \ \mathbb{E}_{\boldsymbol{x},\boldsymbol{y}}\|f(\boldsymbol{y}) - \boldsymbol{x}\|_2^2 \\ f^{\mathrm{CED}} = \arg\min_{f \in \mathcal{S}_{\mathrm{CED}}^c} \mathbb{E}_{\boldsymbol{y}}\|f(\boldsymbol{y}) - \boldsymbol{y}\|_2^2 \\ f^{\mathrm{BS}} = \arg\min_{f \in \mathcal{S}_{\mathrm{BS}}^c} \mathbb{E}_{\boldsymbol{y}}\|f(\boldsymbol{y}) - \boldsymbol{y}\|_2^2 \end{cases}. \tag{63}$$

Standard calculus rules give:

$$\nabla_{\boldsymbol{y}}\log p_{\boldsymbol{y}}(\boldsymbol{y}) = \frac{1}{\sigma^2}\left(\tanh\left(\frac{y_1 + y_2}{\sigma^2}\right)\boldsymbol{1} - \boldsymbol{y}\right), \tag{64}$$

$$f^{\mathrm{MMSE}}(\boldsymbol{y}) = \boldsymbol{y} + \sigma^2\nabla_{\boldsymbol{y}}\log p_{\boldsymbol{y}}(\boldsymbol{y}) = \tanh\left(\frac{y_1 + y_2}{\sigma^2}\right)\boldsymbol{1}, \tag{65}$$

$$f^{\mathrm{CED}}(\boldsymbol{y}) = c\boldsymbol{y} + (1-c)\left(\boldsymbol{y} + \hat{\eta}\nabla_{\boldsymbol{y}}\log p_{\boldsymbol{y}}(\boldsymbol{y})\right) = \boldsymbol{y} + \frac{(1-c)\hat{\eta}}{\sigma^2}\left(\tanh\left(\frac{y_1 + y_2}{\sigma^2}\right)\boldsymbol{1} - \boldsymbol{y}\right), \tag{66}$$

and

$$f^{\mathrm{BS}}(\boldsymbol{y}) = c\boldsymbol{y} + (1-c)\begin{pmatrix}\mathbb{E}\{x_1|y_2\} \\ \mathbb{E}\{x_2|y_1\}\end{pmatrix} = c\boldsymbol{y} + (1-c)\begin{pmatrix}\tanh(y_2/\sigma^2) \\ \tanh(y_1/\sigma^2)\end{pmatrix}, \tag{67}$$

with $\hat{\eta} = (\mathbb{E}_{\boldsymbol{y}}\frac{1}{2}\|\nabla\log p_{\boldsymbol{y}}(\boldsymbol{y})\|_2^2)^{-1}$.

For $f^{\mathrm{DC}} = \arg\min_{f \in \mathcal{S}_{\mathrm{DC}}^c} \mathbb{E}_{\boldsymbol{y}}\|f(\boldsymbol{y}) - \boldsymbol{y}\|_2^2$, there exists no closed-form solution, to the best of our knowledge. However, using Theorem 1, we can train a neural network of the form $\boldsymbol{A}\nabla\psi_{\boldsymbol{\theta}}$ where $\boldsymbol{A} = \begin{pmatrix} 0 & 1 \\ -1 & 0 \end{pmatrix}$ and $\psi_{\boldsymbol{\theta}} : \mathbb{R}^2 \to \mathbb{R}$, to solve

$$\boldsymbol{\theta}^* = \arg\min_{\boldsymbol{\theta}} \ \mathbb{E}_{\boldsymbol{y}}\|\boldsymbol{A}\nabla\psi_{\boldsymbol{\theta}}(\boldsymbol{y}) - \boldsymbol{y}\|_2^2, \tag{68}$$

so that $f^{\mathrm{DC}}$ is well approximated by $\boldsymbol{y} \mapsto c\boldsymbol{y} + (1-c)\boldsymbol{A}\nabla\psi_{\boldsymbol{\theta}^*}$. In practice, we use a simple Multi-Layer Perceptron (MLP) to parameterize $\psi_{\boldsymbol{\theta}}$.

The mean squared error (MSE) results, obtained by simulation, are reported in Table 5 and Figure 3.

## E.2  GAUSSIAN AND SPIKE & SLAB

We do not detail the derivations of the closed-form solutions but the mean squared error (MSE) results are reported in Table 5.

---

[2]In the one-dimensional case, we have $\mathcal{S}_{\mathrm{DC}}^c = \mathcal{S}_{\mathrm{BS}}^c$, which prevents us from highlighting the differences between the two estimators.

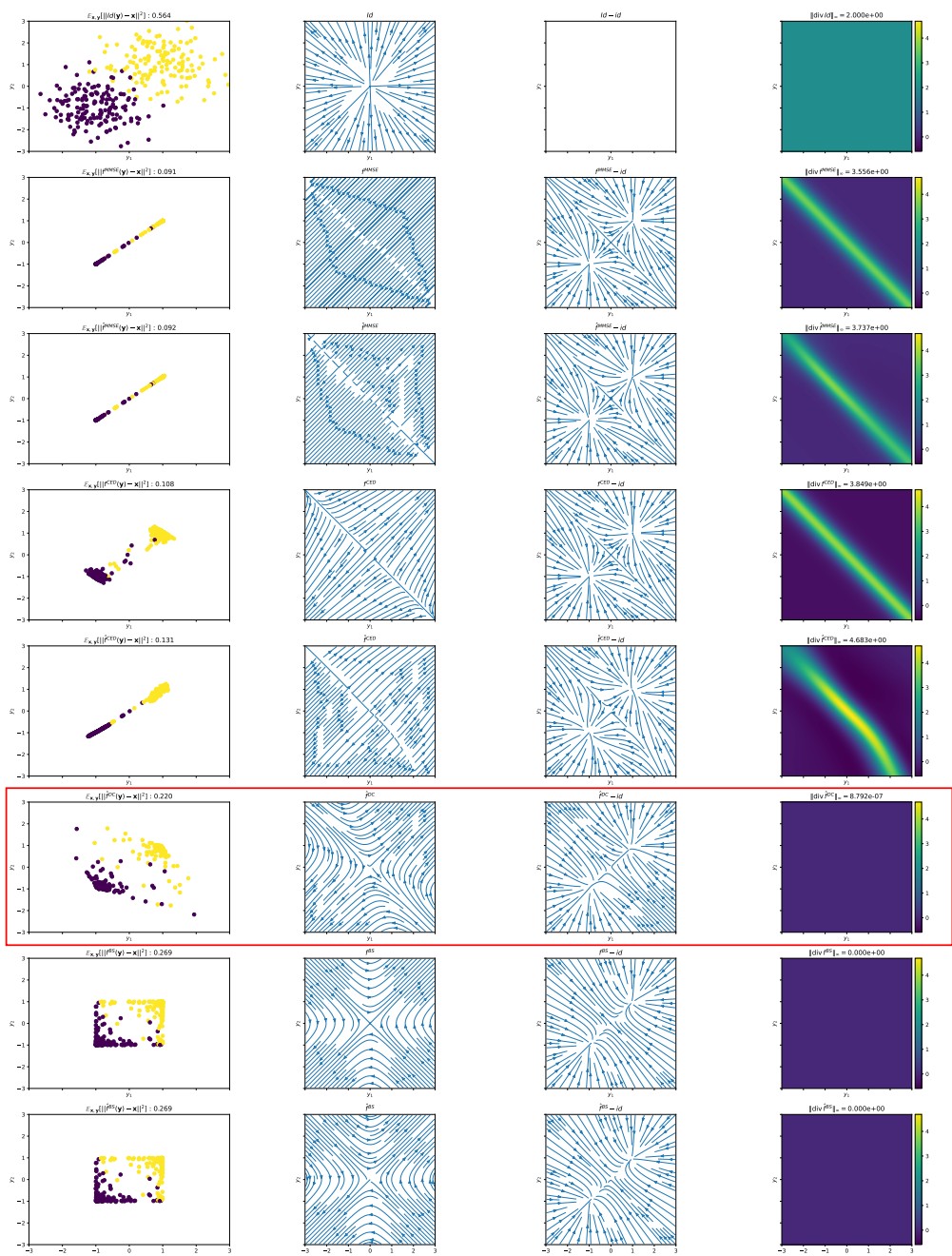

Figure 3: Comparison between MMSE, CED, DC and BS estimators (for $c = 0$, unknown $\sigma$ setting) for the toy signal distribution "Two Deltas". Each row correspond to a different denoiser (from top to bottom: id, $f^{\text{MMSE}}$, $\hat{f}^{\text{MMSE}}$, $f^{\text{CED}}$, $\hat{f}^{\text{CED}}$, $\hat{f}^{\text{DC}}$ (ours), $f^{\text{BS}}$, $\hat{f}^{\text{BS}}$). The denoisers $f^{\text{MMSE}}$, $f^{\text{CED}}$ and $f^{\text{BS}}$ are the closed-form solutions, while $\hat{f}^{\text{MMSE}}$, $\hat{f}^{\text{CED}}$, $\hat{f}^{\text{DC}}$ and $\hat{f}^{\text{BS}}$ refer to the approximated solutions learned from data leveraging a Multi-Layer Perceptron (MLP). Columns show, from left to right: (i) denoised samples $\hat{\boldsymbol{x}}(\boldsymbol{y})$, with the mean squared error $\mathbb{E}_{\boldsymbol{x},\boldsymbol{y}}\|\hat{\boldsymbol{x}}(\boldsymbol{y}) - \boldsymbol{x}\|^2$ reported in the title and computed over the test dataset; (ii) the vector field induced by the denoiser $\boldsymbol{y} \mapsto \hat{\boldsymbol{x}}(\boldsymbol{y})$; (iii) the difference between this field and the identity function; and (iv) the divergence of the denoiser-induced field. Note that $\hat{f}^{\text{DC}}$ and $f^{\text{BS}}$ yield vector fields with identically zero divergence, as visible in the last column. The unknown constant noise level $\sigma$ is set to 0.75.

|  | **Two Deltas** | **Gaussian** | **Spike & Slab** |
|---|---|---|---|
| $p_{\boldsymbol{x}}$ | $\frac{1}{2}\delta_{-1} + \frac{1}{2}\delta_1$ | $\mathcal{N}(\mathbf{0}, \boldsymbol{I})$ | $\frac{1}{2}\mathcal{N}(\mathbf{0}, \boldsymbol{I}) + \frac{1}{2}\delta_{\mathbf{0}}$ |
| $p_{\boldsymbol{y}}$ | $\frac{1}{2}\mathcal{N}(-\mathbf{1}, \sigma^2\boldsymbol{I}) + \frac{1}{2}\mathcal{N}(\mathbf{1}, \sigma^2\boldsymbol{I})$ | $\mathcal{N}(\mathbf{0}, (1+\sigma^2)\boldsymbol{I})$ | $\frac{1}{2}\mathcal{N}(\mathbf{0}, (1+\sigma^2)\boldsymbol{I}) + \frac{1}{2}\mathcal{N}(\mathbf{0}, \sigma^2\boldsymbol{I})$ |
| MMSE | **0.089** | **0.360** | **0.243** |
| MSE CED | 0.106/**0.089** | 1.000/**0.360** | 0.427/**0.243** |
| MSE DC | 0.218/0.157 | 1.000/**0.360** | 0.500/0.265 |
| MSE BS | 0.266/0.181 | 1.000/**0.360** | 0.500/0.265 |

Table 5: Toy signal distributions and their mean squared error (MSE) results for different estimators (left: $c = 0$, right: optimal $c$). The constant noise level $\sigma$ is set to $0.75$.

# F ADDITIONAL EXPERIMENTS

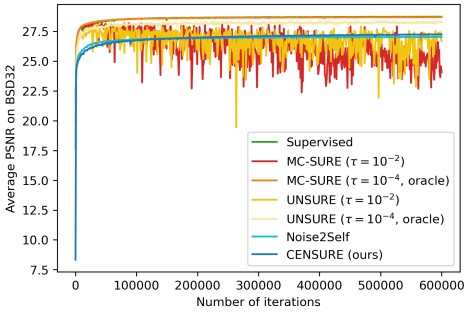

Figure 4: Stability issues during training for Monte Carlo approximation methods (see equation (6)) for grayscale images with constant $\sigma = 25$.

Figure 5: Ablation study on the number of terms $K'$ in CENSURE. Average PSNR (in dB) results on Set12 (grayscale) with unknown constant noise level $\sigma = 25$ is reported for each model. As expected, performance improves as the number of terms increases.

| $K'$ | 2 | 4 | 6 | 8 |
|---|---|---|---|---|
| Set12 | 29.44 | 29.67 | 29.74 | 29.81 |

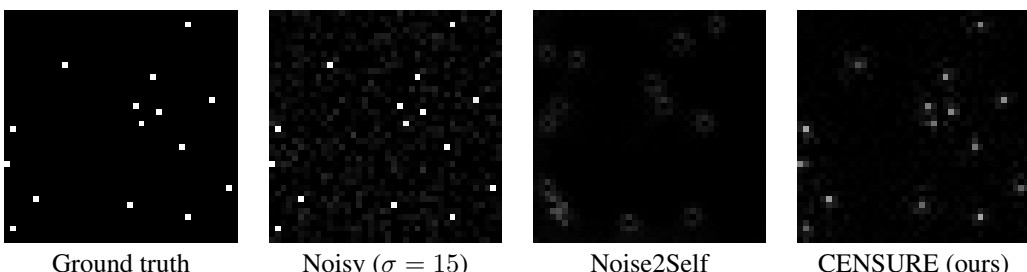

| Ground truth | Noisy ($\sigma = 15$) | Noise2Self | CENSURE (ours) |
|---|---|---|---|

Figure 6: Divergence-free estimators are more expressive than their blind-spot counterparts. Blind-spot networks, by construction, enforce their receptive field to exclude the center pixel $y_i$ when estimating $x_i$, since the component function $f_i$ cannot depend on $y_i$ by definition. In contrast, divergence-free networks can exploit the full image context without masking, including the central pixel $y_i$, which is typically highly informative about $x_i$. It is illustrated here as Noise2Self removes all isolated white data points, whereas a divergence-free estimator can preserve them more accurately, leading to improved denoising performance.

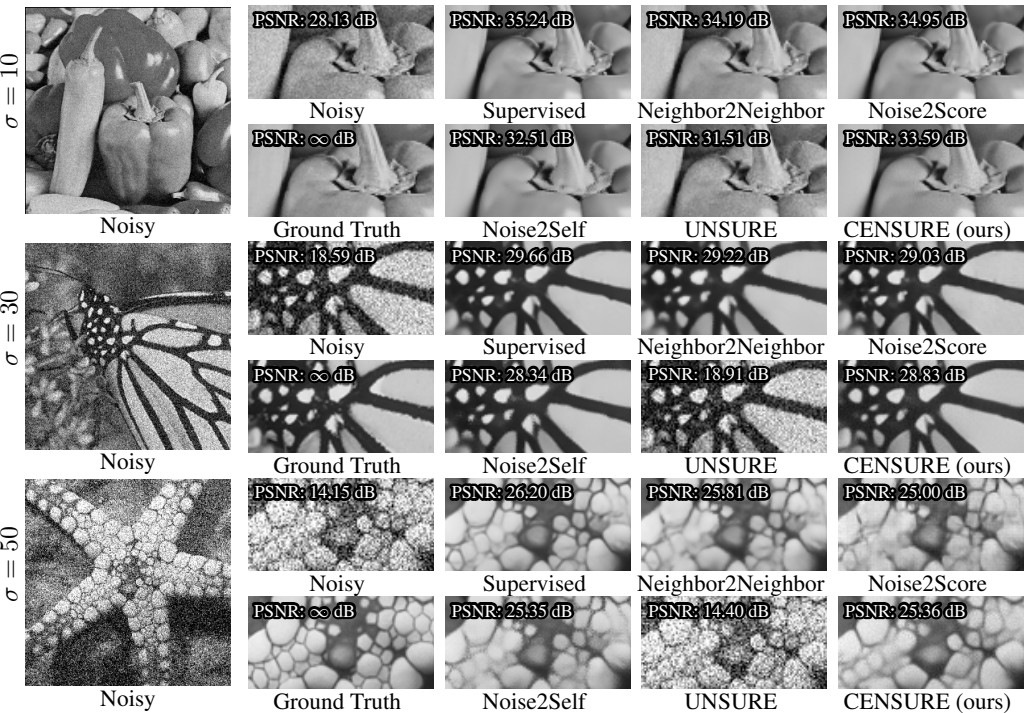

Figure 7: Qualitative blind image denoising results. All models were trained on a dataset with **unknown random** $\sigma \in [0, 50]$ (a single model to handle all noise levels). Best viewed by zooming.

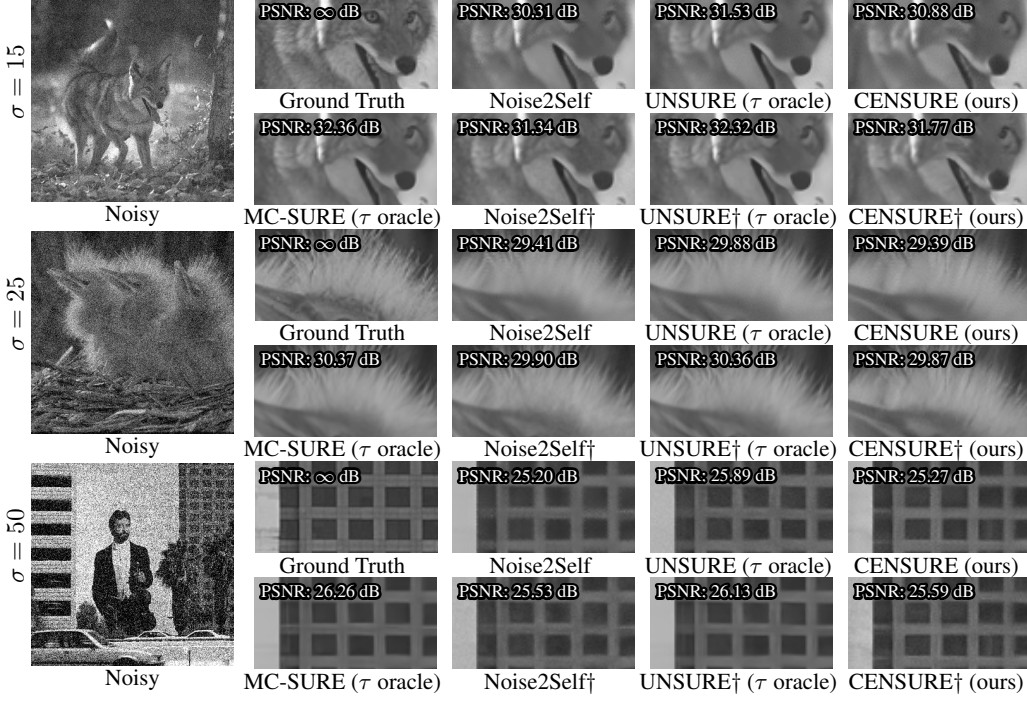

Figure 8: Qualitative image denoising results. All models were trained on a dataset with **constant** noise level $\sigma$ (one model per noise level). Models converted into noise-level–aware variants, using Propositions 1 and 2, are indicated by the symbol † (no additional training). Best viewed by zooming.

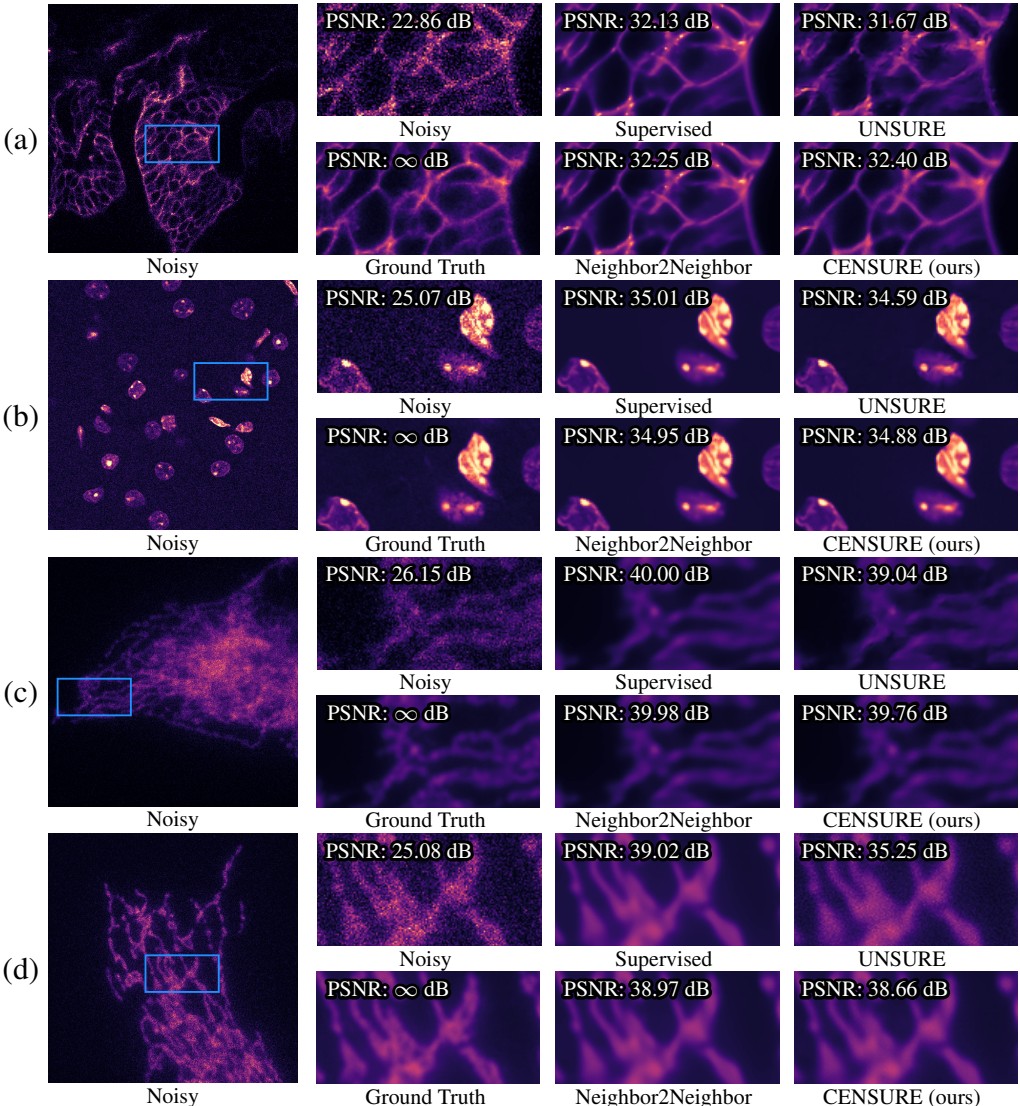

Figure 9: Demonstration of the applicability of self-supervised Gaussian denoisers trained on synthetic data (with constant noise level $\sigma = 25$) to real-world fluorescence microscopy images, using the Noise2VST framework (Herbreteau & Unser, 2025). (a) Confocal fish from the FMD dataset (Zhang et al., 2019); (b) two-photon mice from the FMD dataset; (c–d) fluorescence microscopy images from the W2S dataset (Zhou et al., 2020). The denoisers are the same as those reported in Table 4 under the unknown-noise setting ($\sigma = 25$). Colors are for aesthetic purposes only. Best view by zooming.

