# OpenReview forum: "Divergence-Free Neural Networks with Application to Image Denoising"
_ICLR.cc/2026/Conference — ICLR 2026 Poster_

### Official Review · Reviewer_o8EU · 2025-10-30

**Soundness:** 3
**Presentation:** 3
**Contribution:** 3
**Rating:** 6
**Confidence:** 3

**Summary:**

This paper introduces a theoretically grounded framework for constructing neural networks that are divergence-free for image denoising. The authors propose a resource-efficient parameterization that represents the network output as a structured combination of conservative fields. The proposed divergence-free property is used to simplify SURE loss, avoiding the instability of Monte Carlo approximations and the expressivity limitations of blind-spot methods. The experimental results show that the proposed method achieves better performance against other divergence-based approaches

**Strengths:**

+ The main strength of this paper is its rigorous mathematical analysis.
+ The method for constructing the sparse, parameterized skew-symmetric matrices and sharing the scalar potential network is reasonable. It successfully makes the theoretical representer theorem computationally tractable for high-dimensional data like images.

**Weaknesses:**

- The experimental evaluation is focused only on divergence-based self-supervised methods (MC-SURE, Noise2Self, UNSURE). The field of self-supervised (or unsupervised single-image) denoising is much broader and has developed significantly. The paper is missing crucial comparisons to other state-of-the-art methods that are not based on SURE, which makes it impossible to assess the true competitiveness of the proposed approach, e.g., [1-3]
[1] High-quality self-supervised deep image denoising, NIPS 2019
[2] Iterative denoiser and noise estimator for self-supervised image denoising, ICCV 2023
[3] Positive2Negative: Breaking the Information-Lossy Barrier in Self-Supervised Single Image Denoising, CVPR 2025
- The entire framework is demonstrated only for additive white Gaussian noise. While this is a standard setting for theoretical exploration, its practical use is unexplored. The paper itself acknowledges this limitation but does not provide any discussion or preliminary results on how the divergence-free constraint might perform with more complex, realistic noise models (e.g., Poisson-Gaussian, spatially correlated, or signal-dependent noise). The heavy reliance on SURE makes the direct extension non-trivial
- The proposed parameterization is complex, though more efficient than the full model. This significant overhead could be a barrier to practical adoption, and this trade-off is not sufficiently emphasized in the results

**Questions:**

- The paper argues that the proposed method is more expressive than blind-spot methods because it doesn't ignore the central pixel. However, blind-spot methods are typically blind to the noise distribution, whereas DivFree's theoretical advantage requires knowing the noise level. Could you discuss this trade-off between expressivity and the need for prior information? How does your method perform if the provided noise level is not very precise

---

> ### Author Response · Authors · 2025-11-21
>
> **Comparisons:** Please refer to the general response to all reviewers. Note that we made comparisons only with the methods Neighbor2Neighbor and Noise2Score, recommended by Reviewer 7HQs, since evaluating all suggested methods would have been too time-consuming. However, Noise2Score is close to [1] (known $\sigma$ setting) and Neighbor2Neighbor is close to [2] (unknown $\sigma$ setting). Moreover, [3] is not strictly a self-supervised method since the parameters of the network need to be initialized with parameters learned supervisedly for the removal of Gaussian noise.
>
> **Gaussian noise setting only:** Please refer to the general response to all reviewers.
>
> **Overhead:** We have added a runtime comparison in the Appendix in the Limitations section (see Table 3).
>
> **Knowledge of the noise level:** We believe there may have been a slight misunderstanding: our approach requires knowing that the noise distribution is Gaussian but **not the noise level** $\sigma$, contrary to standard SURE. Indeed, the noise level totally disappears from the loss function when imposing $\text{div}f=0$ (see Eq (11)). However, if the noise level is provided at inference, we can leverage this information to improve the performance, in view of Proposition 1 and 2. We have clarified this in the revised text and included Figure 1 to provide a concise overview.
>
> Our approach can also be used for **blind estimation**. Specifically, training can be conducted using only noisy images corrupted with Gaussian noise and **random** $\sigma \in [0, 50]$. The resulting model ultimately learns, in a self-supervised way, to estimate the noise level and to denoise accordingly. The results of this experiment are presented in Table 1. Note that **UNSURE is ineffective in this setting**, as discussed with Reviewer 7HQs, because the softhen constraint $\mathbb{E} (\text{div}f(\boldsymbol{y})) = 0$ is insufficient to eliminate the dependence on the knowledge of $\sigma$. **This further underscores the strength of our approach**.
>
> However, we agree that there does exist a trade-off between expressivity and the need for prior information, as discussed at the end of Subsection 3.1 and illustrated by the newly added Figure 1.

---

### Official Review · Reviewer_zsZU · 2025-10-31

**Soundness:** 2
**Presentation:** 3
**Contribution:** 2
**Rating:** 6
**Confidence:** 3

**Summary:**

SURE needs divergence. To get rid of the divergence, they introduce a constant divergence field S_DC (Eqn 10). Section 4 is about how to generate such a divergence field. The core trick is by means of a universal approximation (Theorem 1). Realization is given in Section 4.2.

**Strengths:**

Solid contribution.

The coverage of the prior work is excellent.

Intuition of the field construction can be done better. But basically I get the idea that they want to ensure symmetry so that the field is valid (Eqn. 13 and Eqn 14).

Empirical results are okay. This is a piece of theoretical contribution.

**Weaknesses:**

This is not a criticism but a general comment. Will not contribute to my judgment of the paper.

SURE is a proxy for MSE, which assumes that the underlying noise is iid Gaussian. But if I know that the noise is iid Gaussian, I can literally just simulate training data and train my model in a supervised way. The cost of doing that is null. Even if we say that the noise level is unknown, worst case I just train a larger model with more data augmentation. So I have a hard time to convince myself that SURE is the right way to go.

I can appreciate unsupervised learning, but it must be some bizarre unknown noise type that I cannot easily calibrate and know from my camera. But for this case, SURE doesn't seem to be the right approach to go.

So I am not sure about the utility down the road. Perhaps denoising is not a good showcase?? No need to address this in the rebuttal. Just some thoughts to share with the authors.

**Questions:**

No major concerns.

---

> ### Author Response · Authors · 2025-11-21
>
> Thank you for your positive feedback.
>
> **SURE raison d'être:** We fully agree that, when it is possible to collect noise-free data representative of the target application, training a denoiser in a supervised manner is indeed the preferred approach. However, in some applications, it can be particularly challenging to obtain such clean data, making it necessary to train using only noisy observations. This is precisely where SURE and other related unbiased risk estimators come into play. However, when the noise distribution is totally unknown, SURE may indeed fail and more general approaches such as Noise2Self (which only assumes zero-mean and spatially-independent noise) are more recommended (see discussion at the end of Subsection 3.1). We also have added Figure 1 to the revised paper that illustrates the expressivity trade-off that emerges in self-supervised denoising.

---

> > ### Comment · Reviewer_zsZU · 2025-11-21
> > **Thanks.**
> >
> > Thanks for taking the time to share your thoughts. As I mentioned in my original review, the paper is in reasonably good shape with decent contributions. I hope other reviewers can support your paper. Best of luck.

---

### Official Review · Reviewer_G54J · 2025-10-31

**Soundness:** 4
**Presentation:** 4
**Contribution:** 2
**Rating:** 4
**Confidence:** 4

**Summary:**

This paper proposes a new class of divergence-free denoisers by interpolating the constraints of two prior classes of divergence-free denoisers and also a method for constructing denoisers that belong to the new class directly, without the need to use additional loss functions during training like prior methods. The authors show improved performance with known noise levels, and claim this is due to closer adherence to the divergence-free constraint, as well as comparable performance in the unknown noise level setting.

**Strengths:**

The idea of examining a class of objects (in this case, denoisers) in between two other existing classes in the literature is very well motivated. I could not find any issues with the theory, which is built on existing solid results in a reasonable way. The results also seem consistent with the understanding of the method presented in the paper, namely that in the known noise-level regime their more constrained method achieves higher performance than the less constrained baseline which does not enforce zero divergence strictly.

**Weaknesses:**

I'm not sure about the resource efficiency claims. It seems that mostly the case rests on using far fewer terms than necessary as a basis to represent the divergence-free denoiser. I think the paper would be stronger if the authors could show what the exact influence of the number of components, K', is on the denoising performance and how close their denoiser is to being truly divergence free. My understanding of the theory is that it only guarantees a divergence-free field if the number of basis terms is n(n+1)/2, where n is input size, but far fewer basis terms are used for the sake of computational tractability. Additionally, the authors should compare the runtime of the various methods, since it appears that their denoiser requires effectively K'+1 network evaluations, combining forward and backward passes.

**Questions:**

How does the following depend on K'?
- denoising performance
- runtime
- divergence

I would be willing to upgrade my score if the authors could provide a more thorough empirical validation of their method, in particular with respect to the major approximation that causes deviations from theory, the number of basis components of their divergence-free denoiser field.

---

> ### Author Response · Authors · 2025-11-21
>
> **Influence of K'**: We believe there may have been a slight misunderstanding regarding the influence of $K'$ on the divergence of the resulting network. In fact, a direct consequence of Lemma 2 is that the function defined in equation (12) is **exactly** divergence-free for any number of terms $K'$, since the sum of divergence-free functions remains divergence-free. The parameter $K'$ only affects expressivity: when $K'=0$, we have $f=0$, which is indeed divergence-free but not expressive at all; conversely, $K'=K$ yields maximal expressivity, as stated in Theorem 1, but becomes computationally intractable. To obtain a resource-efficient yet expressive divergence-free neural network, we propose selecting $K'$ such that $0<K'\ll K$. In summary, the theory ensures that our network is exactly divergence-free, and we have confirmed this experimentally using a Monte Carlo approximation (see Equation 7). We have clarified this point in the revised paper.
>
> Increasing $K'$ increases expressivity (and so denoising performance) as well as runtime. In the paper, we arbitrarily set $K'=8$. As recommended, we haved trained divergence-free networks with $K'=2, K'=4$ and $K'=6$ in addition to $K'=8$ and compared their denoising performance and runtime. The results are presented in the Appendix in Table 3, which also includes a comparison with the supervised and divergence-based self-supervised counterparts, as recommended. As already mentioned in the Limitations section, our method does incur a higher computational cost during both training and inference. In particular, the overall inference cost corresponds indeed to approximately $K'+1$ network evaluations.

---

### Official Review · Reviewer_7HQs · 2025-11-03

**Soundness:** 3
**Presentation:** 2
**Contribution:** 1
**Rating:** 2
**Confidence:** 5

**Summary:**

This paper presents a **Divergence-Free Neural Network (DivFree)** designed for self-supervised Gaussian image denoising.
The network is parameterized with skew-symmetric bases so that ∇·f(x)=0 holds exactly, thereby eliminating the need for Monte-Carlo divergence estimation used in SURE-based losses.
The authors claim this leads to more stable training and improved performance compared to MC-SURE and UNSURE.
Experiments are limited to grayscale Gaussian noise (σ=15–50), with an additional “unknown σ” case that is not truly blind.

**Strengths:**

- Mathematically consistent derivation of divergence-free vector fields using a skew-symmetric representation.
- Conceptually simple idea that connects physical conservation laws with SURE-based denoising.
- Reduces gradient variance by removing stochastic divergence estimation.

**Weaknesses:**

- **Misleading “unknown σ” claim.** The model is trained with a fixed σ and only tested on slightly different noise levels; there is no random-σ training or blind estimation, so it cannot handle unknown noise.
- **Extremely narrow scope.** All experiments use grayscale Gaussian noise. There are no color image... The method is not validated on **Poisson**, **Gamma**, or real-world datasets, although its principle directly relates to unbiased risk estimators like **PURE (Poisson Unbiased Risk Estimator)**.
- **High computational cost.** The paper itself states that inference is about **9× slower** than DRUNet due to repeated gradient evaluations for enforcing divergence-free constraints.
- **Limited novelty.** The method merely re-parameterizes the denoising backbone to set div(f)=0, optimizing the same SURE objective as MC-SURE. The observed differences stem from reduced variance but increased structural bias.
- **Lack of generality.** Divergence-free constraints could naturally apply to flow, optical-flow, or magnetic-field problems, yet the paper confines itself to toy Gaussian denoising.

## Minor
- No visualization verifying that learned vector fields are actually divergence-free.
- Bias–variance trade-off introduced by the constraint is never quantified.
- Comparison baselines (Noise2Score, Blind2Unblind, Neighbor2Neighbor, diffusion-based denoisers) are missing.

**Questions:**

1. How is the “unknown σ” setting defined? Was σ randomly sampled during training or fixed to one value?
2. Could the same architecture be evaluated on **Poisson denoising** using **PURE** to demonstrate generality beyond Gaussian SURE?
3. Why not apply divergence-free constraints to other tasks (e.g., optical flow, physics-based fields) where this property is physically meaningful?
4. Does enforcing div(f)=0 introduce measurable bias compared to the true SURE optimum?
5. Given the reported 9× inference cost, is the stability improvement practically worthwhile?

---

> ### Author Response · Authors · 2025-11-21
>
> **Major**
> 1. **“unknown σ” claim**: We believe that there might have been a misunderstanding regarding the term “unknown $\sigma$”: it has to be understood the same way as in UNSURE paper. Specifically, training is conducted using only noisy images corrupted with Gaussian noise at **fixed** $\sigma$ but the value of $\sigma$ (assumed unknown) is not used in the loss function (contrary to MC-SURE for example) or during inference/test time (contrary to the models marked with $\dagger$). We have clarified this point in the revised manuscript.
> 2. **Blind estimation**: Thank you for suggesting that we address the blind estimation setting. We overlooked this point in the initial submission, but our approach offers a key advantage over UNSURE: it can naturally accommodate training with random $\sigma$. Indeed, theory allows it, in accordance with (4), as long as the estimator $f$ is divergence-free everywhere (*i.e.*, $\forall \boldsymbol{y} \in \mathbb{R}^n, \text{div}f(\boldsymbol{y}) = 0$). This holds for Noise2Self and our proposed CENSURE but not for UNSURE for which the estimator $f$ is only divergence-free in expectation (*i.e.*, $\mathbb{E}\text{div}f(\boldsymbol{y}) = 0$). Indeed, for UNSURE with non-constant $\sigma$, $\mathbb{E}(\sigma^2 \text{div}f(\boldsymbol{y})) \neq \mathbb{E}(\sigma^2) \mathbb{E}(\text{div}f(\boldsymbol{y})) = 0$ in full generality because $\sigma^2$ is not a constant and is not independent of $\text{div}f(\boldsymbol{y})$, since $\boldsymbol{y}$ itself depends on $\sigma$. Therefore, UNSURE cannot be used, a priori, under a random- $\sigma$ training contrary to our approach. We have evaluated all the divergence-based models under this setting. Specifically, training is conducted using only noisy images corrupted with Gaussian noise and **random** $\sigma \in [0, 50]$. Once again, the values of $\sigma$ (assumed unknown) are not used in the loss function or during inference/test time. The results are presented in Table 1. In this setting, **CENSURE outperforms its divergence-based counterparts** and we believe that **this significantly strengthens our contribution.** We have revised the paper to emphasize this important point throughout, and we thank you again for bringing it to our attention.
> 3. **Narrow scope**: We focused on grayscale images since extending the method to color images is trivial (the number of input and output channels of all networks just change from 1 to 3). Moreover, working with grayscale images is often considered more challenging, as one cannot rely on cross-channel redundancy to aid the estimation. If the paper is accepted and time permits, we will include results on color images as well. As mentioned in the limitation section, our application in image denoising targets only additive white Gaussian noise since Stein’s unbiased risk estimator (SURE) involves only a term of divergence, contrary to the unbiased risk estimators dedicated to other types of noise. For example (see UNSURE), for Poisson noise, PURE involves a term of the form $\sum_{i} y_i \frac{\partial f_i}{\partial y_i}(\boldsymbol{y})$. Designing architectures so that $\forall \boldsymbol{y} \in \mathbb{R}^n, \sum_{i} y_i \frac{\partial f_i}{\partial y_i}(\boldsymbol{y}) = 0$ holds is beyond the scope of this paper.
> 4. **Lack of generality**: Please refer to the general response to all reviewers.
>
> **Minor**
> 1. **Visualization** We have added in Appendix D (see Figure 3) a visualization of the learned vector fields on 2D toy distributions. These experiments show that there are indeed divergence-free. Moreover, in the nD case, we can confirm that the learned vector fields are also divergence-free experimentally using a Monte Carlo approximation (see Equation 7) as it was done in the code provided.
> 2. **Bias:** Enforcing $\text{div}f=0$ does introduce a bias compared to the true SURE optimum. We can derive a lower bound on the reconstruction error using the fact that $S_{DC}^{c} \subset S_{CED}^{c}$. We have added Proposition 3 to the revised paper regarding this point (proof in the Appendix C).
> 3. **Baselines:** Please refer to the general response to all reviewers.
>
> **Questions:**
> 1. Please refer to our response regarding “unknown $\sigma$” and "blind estimation" in the Major section above.
> 2.	The same architecture cannot be used to target Poisson noise unfortunately. Indeed, as discussed in the Major section, PURE involves an additional term of the form $\sum_{i} y_i \frac{\partial f_i}{\partial y_i}(\boldsymbol{y})$ and our architecture only ensures that $\sum_{i} \frac{\partial f_i}{\partial y_i}(\boldsymbol{y})= 0$.
> 3.	Please refer to the general response to all reviewers.
> 4.	Please refer to our response regarding "bias" in the Minor section above.
> 5.	We agree that there exists a trade-off between stability and inference cost. Given that computational resources keep improving year after year, we believe that this trade-off can only shift in our favor in the future.

---

> > ### Comment · Reviewer_7HQs · 2025-11-27
> >
> > Dear Authors,
> >
> > Thank you for your detailed response. Some of my concerns have been addressed, but I still have reservations about the practical usefulness of the proposed methods.
> >
> > 1. You state that Gaussian denoisers are strong baselines for real-world scenarios, but no empirical evidence is provided; the justification remains purely conceptual. You also argue that extending your approach to color-scale denoising is trivial. If this extension is indeed straightforward, then you should demonstrate experimentally that your proposed method actually works in such realistic settings. In its current form, it is difficult to see how these methods can be applied in practice, and their applicability to other tasks is still unclear.
> >
> >
> > 2. There also appear to be some inaccuracies regarding existing baselines. Noise2Score is applicable to unknown noise levels (unknown σ), and the extended version, Noise2Adapt2Score, is truly designed for blind image denoising, where both the noise level and distribution are unknown and must be predicted. Please correct these descriptions accordingly.
> >
> > Noise2Adapt2Score [1] was proposed as a CVPR 2022 method specifically for this blind setting.
> >
> >
> >
> >
> > Finally, if you could clearly highlight the revised parts in the manuscript, it would be very helpful for reviewers to track the changes.
> >
> > [1] K. Kim et al., “Noise Distribution Adaptive Self-Supervised Image Denoising using Tweedie Distribution and Score Matching,” CVPR, 2022.

---

> > > ### Author Response · Authors · 2025-11-27
> > >
> > > Dear Reviewer,
> > >
> > > Thank you very much for your feedback. As recommended, we have highlighted in red all the changes to the manuscript in order to ease the review process.
> > >
> > > 1. We have just launched training on color images and we are currently working on the demonstration of our proposed self-supervised Gaussian denoiser for real-world scenarios. We will keep you updated on the results.
> > >
> > > 2. Thanks for the additional reference. We focused on the original paper Noise2Score and were not aware of the extension that you mention. We will correct these descriptions accordingly and cite Noise2Adapt2Score, as recommended.

---

> > > > ### Author Response · Authors · 2025-12-01
> > > >
> > > > Dear Reviewer,
> > > >
> > > > 1. - We have now added a demonstration showing that our proposed divergence-free denoisers can also be trained in a blind, self-supervised fashion for **color images** (see Figure 9 in the Appendix). We have also modified the code provided in the Supplementary Material to include the weights for blind color-image denoising.
> > > >
> > > >    - Regarding real-world applications, we have added a demonstration that all Gaussian denoisers trained for our theoretical study on synthetic data can be directly used for **real-world image denoising**. In Figure 10, we illustrate this by denoising real noisy fluorescence microscopy images from multiple datasets using the Noise2VST framework [a], which requires only an off-the-shelf Gaussian denoiser.
> > > >
> > > >
> > > >    [a] S. Herbreteau and M. Unser. Self-calibrated variance-stabilizing transformations for real-world image denoising. ICCV'25.
> > > >
> > > >
> > > > 2. We have included the extented version of Noise2Score on both Table 1 and 2, as recommended, and corrected the description of this approach accordingly in the manuscript (text highlighted in blue in Section 5).
> > > >
> > > >
> > > > Finally, we would like to sincerely thank you. Without your critical yet always constructive feedback, the paper would not have reached its current form.

---

### Author Response · Authors · 2025-11-21
**Common response to all reviewers**

First of all, we would like to express our sincere gratitude to the reviewers for dedicating their valuable time to the evaluation of our paper and for providing us with constructive feedback. We thank the reviewers for highlighting the **rigorous mathematical analysis** (Reviewers o8EU and 7HQs) of our paper, noting its **excellent soundness and presentation** (Reviewer G54J), and recognizing it as a **solid piece of theoretical contribution** with **excellent coverage of prior work** (Reviewer zsZU). We have revised our paper, taking into account all the comments received.

In particular, **substantial revisions have been made to address the blind estimation setting**, as suggested by Reviewer 7HQs. We had not realized this point in the initial submission, but **our approach actually offers a key advantage over UNSURE: it naturally accommodates training with unknown random noise level $\sigma$** (see Section 5 of the revised manuscript for further details). We believe that **this point significantly strengthens our contribution.** Our approach merits a name that better reflects it and we therefore rename DivFree to CENSURE (Concealed and Erratic Noise level with Stein’s Unbiased Risk Estimate). We have also included Figure 1 to more clearly situate our contribution within the broader context.

Criticism from some reviewers focused on the application domain, namely denoising, while our approach could have been applied to other research fields related to physics-informed machine learning such as flow, optical-flow, or magnetic-field problems, where enforcing incompressibility of the estimator is often important in accordance with the fundamental physical laws. We chose denoising as our application mainly because the authors of this paper have expertise in image processing and because our approach fills a gap in self-supervised learning. Moreover, denoising is often considered as a testbed: it is one of the simplest high-dimensional problem but scaling challenges arise quickly. Nevertheless, we agree that extending our method beyond denoising represents an interesting direction for future work.

A few reviewers observed that the present approach focuses exclusively on additive white Gaussian noise. This is because, among unbiased risk estimators for self-supervised denoising, SURE, the estimator for Gaussian noise, is the only one that depends solely on a divergence term, to the best of our knowledge. However, we do not view it as a major weakness as the primary goal of our work is to demonstrate the construction of divergence-free neural networks, with Gaussian denoising serving as an illustrative application. Furthermore, focusing exclusively on Gaussian noise is not particularly restrictive, as recent works [a, b] have shown that deep learning–based Gaussian denoisers could be leveraged in practical real-world scenarios.

Finally, we agree with the reviewers that adding a comparison with non-divergence-based methods was necessary to better assess the competitiveness of the proposed approach. As recommended by Reviewer 7HQs, we added in Table 1 and 2 a comparison with Neighbor2Neighbor (unknown $\sigma$ setting) and Noise2Score (known $\sigma$ setting) as baselines. Note, however, that Neighbor2Neighbor is inherently limited to natural images, as it relies on the core assumption that two noise-free neighboring pixels share similar values most of the time. In contrast, divergence-free approaches are more general for denoising problems. We have also added comparisons with Monte Carlo approximation methods (MC-SURE and UNSURE) for another choice of hyperparameter $\tau$ that we selected based on test-set performance (an oracle hyperparameter), which yielded improved results.

For further details, please refer to the specific responses to each reviewer comment.

[a] S. Herbreteau and M. Unser. Self-calibrated variance-stabilizing transformations for real-world image denoising. ICCV'25.

[b] T. Li et al. Positive2negative: Breaking the information-lossy barrier in self-supervised single image denoising. CVPR'25.

---

### Author Response · Authors · 2025-12-01
**Summary comment for the newly assigned AC**

Dear Area Chair,

We believe a summary comment is necessary to clarify the situation surrounding our paper after the information leakage, which prematurely halted the review process.

**Only 2 out of 4 reviewers** had time to respond to our rebuttal before the sudden interruption:

- Reviewer zsZU had already expressed support for our paper and maintained their positive opinion.

- Reviewer 7HQs was initially the most critical. Following the **substantial revisions** (including the addition of **Figures 1 and 2, Table 1, and a complete rewrite of Section 5**) made during the rebuttal to address their concern regarding **the blind estimation setting**, we believe our contribution has been **significantly strengthened**, particularly as our method now clearly highlights its **key theoretical advantage** over other divergence-based approaches. Reviewer 7HQs noted that “some of [their] concerns have been addressed,” but still had reservations about real-world applications and color-scale denoising. **Both points were resolved** through the newly added **Figures 9 and 10** in the Appendix, though the reviewer did not have the opportunity to acknowledge these updates. In light of the constructive discussion we had with Reviewer 7HQs, we anticipate that their revised score would have reflected the thorough resolution of their concerns.

Unfortunately, the other two reviewers did not have time to respond to our rebuttal:

- Reviewer o8EU had already expressed support for our work. We have addressed their remaining concerns (comparisons, Gaussian noise setting, knowledge of the noise level) primarily in the common response provided to all reviewers.

- Reviewer G54J stated that “[they] would be willing to upgrade [their] score” if we were able to demonstrate that our network is **exactly** divergence-free, which is indeed guaranteed by Lemma 2 (divergence-free functions are closed under addition). We have clarified this point in the revised manuscript (lines 330 to 335). We have also added **Figure 3** in the Appendix to provide numerical evidence that our proposed network maintains strict zero divergence everywhere on a 2D example.

In light of all this, we believe that the reviews and scores, reverted before the discussion period, **no longer reflect the current form of the paper**, which has been **substantially strengthened** thanks to the reviewers’ constructive feedback, for which we are once again sincerely grateful. We respectfully hope that this context will be taken into account.

Wishing you the best of luck with the extra workload brought on by this unfortunate information leakage,

Sincerely,

The Authors

---

### Meta-Review · Area_Chair_PbMr · 2026-01-07

**Summary:**

The paper propose a method that bypass the need to calculate the divergence for image denoising applications. The approach is interesting. It suggests to construct the neural network such that its divergence is zero and thus if estimators such as Stein are used, their divergence term is zero and therefore it is easy to optimize their objective as there is no divergence term. They demonstrate their approach for the task of denoising and show nice results that are robust to knowing the real noise variance.

**Reviewer Concerns:**

There were concerns about comparison to other works, applicability to real world scenarios and questions about the resource efficiency.

**Reviewer Scores:**

It seems that the reviewers answered many of the concerns raised and it is likely that some of the reviewers would increase their scores.

---

### Decision · Program_Chairs · 2026-01-26

Accept (Poster)